

# The importance of terrestrial weathering changes in multimillennial recovery of the global carbon cycle: a two-dimensional perspective

**Marc-Olivier Brault[1,*], H. Damon Matthews[2], and Lawrence A. Mysak[3]**

[1]Department of Geography, McGill University, Montreal, Canada

[2]Department of Geography, Planning and Environment, Concordia University, Montreal, Canada

[3]Department of Atmospheric and Oceanic Sciences, McGill University, Montreal, Canada

*Corresponding Author*: Marc-Olivier Brault, Department of Geography, McGill University

Montreal, Canada H3A 0B9.  E-mail: marc-olivier.brault@mail.mcgill.ca



## 1 Abstract

In this paper, we describe the development and application of a new spatially-explicit weathering
scheme within the University of Victoria Earth System Climate Model (UVic ESCM). We
integrated a dataset of modern-day lithology with a number of previously devised
parameterizations for weathering dependency on temperature, primary productivity, and runoff.
We tested the model with simulations of future carbon cycle perturbations, comparing a number
of emission scenarios and model versions with each other and with zero-dimensional equivalents
of each experiment. Overall, we found that our two-dimensional weathering model versions
were more efficient in restoring the carbon cycle to its pre-industrial state following the pulse
emissions than their zero-dimensional counterparts; however, in either case the effect of this
weathering negative feedback on the global carbon cycle was small on timescales of less than
1000 years. According to model results, the largest contribution to future changes in weathering
rates came from the expansion of tropical and mid-latitude vegetation in grid cells dominated by
weathering-vulnerable rock types, whereas changes in temperature and river runoff had a more
modest direct effect. Our results also confirmed that silicate weathering is the only mechanism
that can lead to a full recovery of the carbon cycle to pre-industrial levels on multi-millennial
timescales.

## 19 Keywords

Weathering; Carbon cycle modeling; Future climate change; Biogeochemistry.



**1    Introduction**
**1.1    Rationale**
The weathering of carbonate and silicate rocks on land is a key process in the global carbon
cycle and, through its coupling with calcium carbonate deposition in the ocean, is the primary
sink of carbon on geologic timescales (Urey, 1952; Walker et al., 1981). The rate at which these
processes remove carbon from the Earth system is sensitive to changes in the environment,
notably temperature (Berner, 1991), biological productivity (Lenton and Britton, 2006) and
perhaps more indirectly, river runoff (Walker and Kasting, 1992). This gives rise to a negative
feedback mechanism which regulates the global climate on multimillennial time scales.
However, there have been but very few quantitative assessments of its impacts on carbon cycling
and ocean biogeochemistry, and its relevance over time frames of $10^4$ years or shorter is largely
unknown.
Here, we introduce a new model of rock weathering developed for use within the University of
Victoria Earth System Climate Model (UVic ESCM); this model incorporates a spatially explicit
interactive computation of weathering rates to close the global carbon cycle on multi-millennial
time scales. The model is based on a lithology-dependent calculation of steady-state weathering
fluxes, which are modulated by transient changes in environmental conditions akin to the 0-D
carbon cycle models already present in the literature (e.g. Meissner et al., 2012). We tested the
model with simulations of future climate changes following anthropogenic carbon emissions,
comparing the output to that of earlier weathering models, both 2-D (Colbourn et al., 2013) and
0-D (Lenton and Britton, 2006).



**1.2    The rock weathering cycle**
The chemical weathering of rocks is characterized by the cleavage of bonds of the mineral lattice
by water, often in the presence of a secondary weathering agent – hydronium or OH ions, low
molecular weight organic chelators, or carbonic acid ($H_2CO_3$; a product of carbon dioxide
dissolution in rainwater).  Rock weathering products, including calcium and bicarbonate ions
(respectively the most abundant cation and anion in most river waters), can be carried away with
runoff to rivers and into the ocean.  For example, calcium carbonate dissolution by carbonic acid
is given by (Archer et al., 1997):
$$CaCO_3 + CO_2 + H_2O \rightarrow Ca^{2+} + 2HCO_3^-$$    (1)
The influx of dissolved inorganic carbon (henceforth DIC) and alkalinity to the ocean surface
layer is balanced by the precipitation and burial of biogenic calcium carbonate ($CaCO_3$) in the
marine sediments, and ocean alkalinity is a key factor in determining the carbonate compensation
depth (CCD), the depth below which the dissolution rate of calcium carbonate exceeds its
precipitation rate.  In the long term this allows the ocean to maintain a remarkably stable
alkalinity, as any increases in ocean acidity (such as can be caused by a $CO_2$ invasion from the
atmosphere) can be neutralized by elevating the CCD, which dissolves carbonate sediments and
releases carbonate ions ($CO_3^{2-}$) back into the ocean.  This oceanic buffer factor, along with
carbonate dissolution on land (due to weathering), is the primary means through which ocean
alkalinity is restored, and is responsible for maintaining both atmospheric and oceanic $pCO_2$
close to equilibrium.  In short, the weathering of calcium carbonate can accelerate the transfer of
$CO_2$ between the atmosphere and ocean, but does not contribute to a permanent return of carbon
to the geologic reservoir (Ridgwell and Zeebe, 2005; Sarmiento and Gruber, 2006).



A certain fraction of rock weathering reactions involve a weakening of chemical bonds in the
mineral lattice on contact with water whereby hydrogen ions replace positively charged cations
(mostly $Ca^{2+}$ and $Mg^{2+}$) which are bounded to negatively charged ions, most particularly $SiO_4$
(silicate) structures.  One of the most common examples is given by calcium silicate hydrolysis,
as described by following schematic reaction (Ebelmen, 1845; Urey, 1952):
$$CaSiO_3 + 2CO_2 + 3H_2O \rightarrow Ca^{2+} + H_4SiO_4 + 2HCO_3^- \qquad (2)$$
This equation represents the weathering of any silicate mineral into silicic acid (which often
precipitates as amorphous silica $SiO_2$), and consumes one more molecule of $CO_2$ than carbonate
dissolution while sending the same amounts of calcium and bicarbonate ions to the ocean.  The
combination of equation (2) with calcium carbonate precipitation (the reverse of equation 1)
shows how this results in a net removal of one molecule of $CO_2$:
$$CaSiO_3 + CO_2 \rightarrow CaCO_3 + SiO_2 \qquad (3)$$
Weathering rates due to silicate hydrolysis tend to be considerably slower than from the
dissolution of carbonate minerals – it removes on average 0.28 to 0.30 Pg C per year (Amiotte
Suchet and Probst, 1995) – hence the effect of atmospheric $CO_2$ consumption by silicate
weathering only becomes a significant sink of carbon on geologic timescales ($10^5$-$10^6$+ years).
For the remainder of this article, dissolution of carbonates (on land) and hydrolysis of silicates
will be treated separately and referred to as carbonate and silicate weathering, respectively.
**1.3    Weathering in early carbon cycle models**
Variations in rock weathering rates have long been believed to hold a major role in regulating the
Earth's long-term climate, and early (non-spatially explicit) carbon cycle models were built to
investigate the importance of the weathering feedback mechanism on various events in Earth's



geological history.  Walker, Hays and Kasting (1981), henceforth referred to as WHAK,
developed expressions relating silicate weathering rates to atmospheric $pCO_2$ (indirectly through
vegetation productivity) and temperature (including a weak dependency on runoff) and used
them to offer a solution to the faint young Sun paradox by providing a convenient mechanism for
a slow and steady decrease in atmospheric greenhouse gas concentrations.  Berner, Lasaga and
Garrels (1983), henceforth referred to as BLAG, also linked the rate of atmospheric $CO_2$
consumption by silicate weathering to changes in surface air temperature, atmospheric partial
$CO_2$ pressure, and river runoff and offered this dependency as a possible explanation for the
general decreasing trend of atmospheric $CO_2$ levels on geologic timescales.  Although these
models used rudimentary parameterizations derived from early general circulation models and
experimental data, they built a foundation for future long-term carbon cycle model studies.
Following BLAG, Berner (1991) built a geochemical cycle model in which the long-term
evolution of atmospheric carbon content would be driven by imbalances between $CO_2$ outgassing
by volcanic activity and the burial of carbonate sediments following the weathering of silicate
rocks.  The latter was given a dependency on air temperature and atmospheric $CO_2$, and it was
used to solve a series of mass balance equations in order to determine the inward and outward
fluxes for the atmosphere-ocean, land, and mineralogical carbon reservoirs.  This new model,
called GEOCARB, added more direct biological mechanisms (notably, the soil-biological
enhancement of weathering) and introduced land elevation and runoff as independent
parameters.  Much like its predecessor, this model was developed in order to reconstruct the
evolution of atmospheric $pCO_2$ over the past hundreds of millions of years.  Subsequent versions
were called GEOCARB II (Berner, 1994) and GEOCARB III (Berner and Kothavala, 2001), and
these further improved the weathering parameterizations based on the latest observational data



and GCM output.  They were later coupled with a model of atmospheric $O_2$ and ocean nutrients
to create COPSE (Bergman et al., 2004), a multi-element geochemical cycling model which
introduces a feedback-based interaction between biotic and abiotic elements of the Earth system.
**1.4    Modelling the lifetime of anthropogenic $CO_2$**
A number of studies have addressed the consequences of anthropogenic carbon emissions and
the multi-millennial lifetime of its perturbation on the climate system.  The earliest attempt to
quantify the timescale of the weathering negative feedback mechanism can be traced back to
Sundquist (1991), who used a coupled atmosphere-ocean-carbon cycle model to obtain an e-
folding timescale on the order of several $10^5$ years.  To this date, it remains the only study to
directly quantify this timescale, although values of 200 kyr (Archer et al., 1997) and 400 kyr
(Berner and Kothavala, 2001; Archer, 2005) have been cited in the literature.  Most recent work
on this matter (Ridgwell and Hargreaves, 2007; Montenegro et al., 2007; Uchikawa and Zeebe,
2008; Archer et al., 2009; Eby et al., 2009) has involved intermediate-complexity models, which
are usually limited in scope to $10^4$ years or shorter.  Hence the weathering feedback mechanism
in these studies is limited to the pH neutralization effect of carbonate weathering on the oceans
(which restores the lysocline to its original depth), and silicate weathering is either ignored
altogether or prescribed as a global constant average flux.
In a pioneering study, Walker and Kasting (1992) considered the impact of the rock cycle and
carbonate sediment dissolution in projections of future changes in the global carbon cycle.  Their
model was built on the assumption that the dependency of carbonate and silicate weathering rates
to changes in the carbon cycle (aka. atmospheric $CO_2$ levels) was purely abiotic, which was in
line with the other geochemical cycling models of the time.  Following on Walker and Kasting



(1992) as well as the recent innovations in COPSE, Lenton and Britton (2006) posited that
biological changes in the Earth system could further enhance the increase or decrease in rock
weathering rates, especially in the context of a rapidly warming world which would likely result
from unabated anthropogenic emissions. Their carbon cycle model included sophisticated biotic
and abiotic transports of carbon, introducing a box-model representation of carbonate and silicate
weathering processes in which weathering rates were directly dependent on plant productivity,
rather than on atmospheric $CO_2$ concentrations. This allowed them to investigate the role of land
use changes on the long-term recovery of atmospheric $CO_2$; in particular, they found that
vegetation-suppressing land use changes would force $CO_2$ levels to stabilize above preindustrial
levels on geologic timescales, thus indefinitely trapping some of the anthropogenic emissions in
the atmosphere.
**1.5    On spatially-explicit weathering models**
Few attempts have been made to explore the spatial variability of carbonate and silicate
weathering rates and how it may affect the global efficacy of the weathering negative feedback
mechanism. The main problematic relating to the development of spatially explicit weathering
schemes is the necessity to compute weathering individually for each land grid cell, which is
entirely more complex than using a globally-averaged value, for which many precedents exist in
the literature. The GEM-$CO_2$ model (Amiotte Suchet and Probst, 1995) addressed this problem
by defining the spatial variability in terms of rock types, and using data for bicarbonate ($HCO_3^-$)
concentration and runoff collected over various mono-lithologic drainage basins (Meybeck,
1987) to establish empirical linear relationships between weathering flux and runoff for a series
of major rock types. Arguing that these two factors (runoff and rock type) were the main factors
controlling the consumption of atmospheric/soil $CO_2$ by weathering, they calculated the global



distribution of $CO_2$ consumption. Their results showed a higher intensity of weathering in the
Northern Hemisphere (due to rock type) and in equatorial regions (due to runoff). They later
refined the global distribution of rock types by attributing one of six rock types to each land unit
of a $1° \times 1°$ grid (Amiotte Suchet et al., 2003); this distribution will be used as the basis for our
spatially explicit weathering scheme. Rock types, from having the smallest to largest impact on
weathering, are classified as follows: plutonic and metamorphic (shield) rocks, sands and
sandstones, extrusive igneous (acid volcanic) rocks, basalts, shales and evaporites, and carbonate
rocks. The latter designs a loose group of predominantly carbonate-based rocks, and of the other
five rock types only sandstones and shales contain a fraction of carbonate-weathered minerals.
In the other rock types, the prevalence of carbonate minerals is too variable and difficult to
estimate, hence they are assumed to contain only silicate-weathered minerals. Using the GEM-
$CO_2$ model, carbonate rocks and shales were found to both consume 40% of the total continental
$CO_2$ uptake despite occupying a much smaller fraction of land area, while sandstones and shield
rocks contributed much lower than their outcrop abundance. A similar rock type distribution
was developed (Gibbs and Kump, 1994; Bluth and Kump, 1994) (hence GKWM), using both an
empirical linear coefficient and an exponential factor to express weathering dependence on
runoff for different rock types; however the results produced by their lithological distribution
was found to be very similar to that of GEM-$CO_2$, in terms of global weathering intensity and the
consumption of atmospheric/soil $CO_2$ (Colbourn et al., 2013).
Other instances of spatially-explicit weathering models are few in the literature. The GEOCLIM
model has a built-in two-dimensional weathering scheme that has been used to investigate the
climatic impacts of tectonic continental reorganization and weathering-vegetation interactions
(Donnadieu et al., 2009). More recently, a spatially-explicit scheme was added to the GENIE



model (Colbourn et al., 2013), using lithological databases from the GEM-CO$_2$ and GKWM
models, temperature dependency from the GEOCARB models, NPP dependency as introduced
by Lenton and Britton (2006), and runoff dependency from GEM-CO$_2$. Although the paper
focused mostly on exploring the various model options, the authors were able to simulate the
entirety of the climate system recovery from a 5000 Pg C anthropogenic pulse at year 2000,
showing that within 0.5-1 Myr the atmospheric CO$_2$ levels would return to pre-industrial levels.
**2      Methods**
**2.1      Climate model description**
In this study we used version 2.9 of the University of Victoria Earth System Climate Model
(henceforth UVic ESCM, or UVic model), which is an intermediate complexity coupled
atmosphere/ocean/sea-ice model with integrated land surface and vegetation schemes (Weaver et
al., 2001). Its main component is version 2.2 of the GFDL Modular Ocean Model (MOM), a
three-dimensional ocean general circulation model with 19 uneven vertical levels (Pacanowski,
1995), which is coupled to a vertically integrated energy-moisture balance atmosphere model
(Fanning and Weaver, 1996), a dynamic-thermodynamic sea-ice model (Bitz et al., 2001), a land
surface scheme and dynamic global vegetation model (Meissner et al., 2003), and a
sedimentation model (Archer, 1996). Land surface properties (surface temperature, soil moisture
content and temperature, and snow cover) and soil carbon content are computed with a single (1-
meter) layer version of the Meteorological Office Surface Exchange Scheme version 2 (MOSES-
2) (Cox et al., 1999), and terrestrial vegetation dynamics are handled by the Hadley Centre's
Top-down Representation of Interactive Foliage and Flora Including Dynamics (TRIFFID)
model (Cox, 2001). TRIFFID describes the state of the terrestrial biosphere in terms of soil



carbon content and vegetation distribution, which is expressed through the structure and
coverage of five plant functional types: broadleaf tree, needleleaf tree, $C_3$ grass, $C_4$ grass, and
shrub vegetation.
The UVic ESCM also includes a fully coupled global carbon cycle, which consists of inorganic
carbon chemistry and land-surface exchanges of $CO_2$ (Ewen et al., 2004), and a Nutrient-
Phytoplankton-Zooplankton-Detritus (NPZD) module which calculates the contribution of the
biological pump to ocean biogeochemistry (Schartau and Oshlies, 2003; Schmittner et al., 2008).
Terrestrial carbon fluxes and reservoirs are described by Matthews et al. (2005), and coupled to
the global model by Meissner et al. (2003).
The model is driven in the short term by seasonal variations in solar insolation and wind fields
(Kalnay et al., 1996), and in the long-term by orbital parameter changes and a reconstruction of
atmospheric $CO_2$ content over the past 20 thousand years (Indermühle et al., 1999).   The spatial
coverage and height of continental ice sheets is prescribed every 1000 years using data from the
model ICE-5G (Peltier, 2004); thus these ice sheet configurations also serve to drive climate
changes during glacial periods.  The land-sea configuration used in all sub-components operates
in a global spatial domain with a spherical grid resolution of 3.6° (zonal) by 1.8° (meridional),
which is comparable to most coupled coarse-resolution AOGCMs.
**2.2     Weathering model description**
Terrestrial weathering in the UVic model is parameterized as a land-to-ocean flux of dissolved
inorganic carbon ($F_{DIC}$) and alkalinity ($F_{ALK}$, with $F_{ALK} = 2F_{DIC}$) via river discharge.   In the
standard version of the model, the incoming flux of carbon to the ocean as weathering is set to
equal the sedimentation rate of $CaCO_3$ in order to balance the long-term carbon and alkalinity



budgets in the ocean; the initial, steady-state value is typically held constant throughout the
transient model runs. This effectively suppresses the long-term negative feedback mechanism by
preventing the weathering rate from adapting to changes in environmental factors such as
temperature and atmospheric $CO_2$ concentration. Meissner et al. (2012) replaced the standard
parameterization of weathering in the UVic model with a number of adaptations from previous
carbon-cycle box models in order to investigate the role of rock weathering as a carbon sink for
anthropogenic carbon emissions. They found that the long-term climate response to various
emission scenarios depends almost exclusively on the total amount of $CO_2$ released regardless of
the rate at which it is being emitted, and carbon uptake through an increase in terrestrial
weathering has a significant effect on the climate system. There were, however, some
differences between the various weathering schemes concerning the rate of carbon removal.
In this section we describe a spatially explicit weathering scheme developed for use within the
UVic ESCM. Steady-state carbonate and silicate weathering rates are calculated for each land
grid cell based on the local rock type (Sect. 2.2.1) and runoff (Sect. 2.2.2). In transient model
runs, these values are modulated by changes in temperature, atmospheric $CO_2$ concentrations or
vegetation productivity, and runoff (Sect. 2.2.3), which are updated on each time step based on
model output. Changes in carbonate and silicate weathering rates are returned to the model in
the form of a riverine flux of carbon and alkalinity (Sect. 2.2.4), which is routed to the ocean.
*2.2.1   Worldwide distribution of rock types*
The two-dimensionality of the weathering model is rooted in the uneven distribution of rock
types across the world. Thus, regions with more active lithologies (mostly sedimentary rocks
such as carbonates and shales) yield higher weathering rates under similar climate conditions,





and these are more sensitive to changes in climate controls than regions predominantly covered by weathering-resistant lithologies (igneous and metamorphic rocks, basalts and granites). Whereas the worldwide distribution of continental rock lithology is well known, there is only limited knowledge of the impact of different rock types on the amounts of riverine exports, therefore any estimation of weathering rates based on local lithological composition is subject to some discrepancy. Two of the most prominent spatially explicit weathering schemes, the Gibbs and Kump weathering model (Gibbs and Kump, 1994), and GEM-CO2 (Amiotte Suchet and Probst, 1995), each use their own set of land lithological data, which classify the entirety of the world's lithologies into one of several rock types, defining the impact of each on weathering by modifying its basic dependency on runoff. While developing a 2-D weathering model for use into the climate model GENIE, Colbourn et al. (2013) compared the output generated by both lithological distributions. They found that the end result did not differ much between the two models (they found a difference of only 4 ppm in atmospheric $CO_2$ concentration 100 kyrs following a pulse of 5000 Pg C in the atmosphere), concluding that differences between individual rock distribution datasets have a negligible impact on the model output.

In this study we used the lithological distribution paradigm first introduced in GEM-CO2, and later published by Amiotte Suchet et al. (2003). The flux of atmospheric/soil CO2 from chemical weathering on each continental grid cell was given an empirical linear relationship to runoff (see Sec. 2.2.2) depending on its assigned predominant rock type, which was classified as one of six different lithological categories: sands and sandstones, shales, carbonate rocks, shield rocks, acid volcanic rocks, and basalts. Sedimentary rocks (limestones, shales, sandstones) contain significant amounts of carbonate rocks, and thus do not consume atmospheric CO2 as efficiently as other rock types, despite sending a higher riverine flux of weathering products.





The adaptation of the rock type distribution map to the UVic model is shown in Figure 1. The
spatial resolution of the UVic model (3.6°×1.8°) is about 6.5 times coarser than that of the
original database (1°×1°) hence the adapted rock distribution paradigm was defined according to
the partitioning of rock types within the area contained by each UVic model grid cell. The
resulting runoff multiplier and carbonate to silicate fractionation therefore becomes a weighted
spatial average of all of rock type multipliers in Table 1.
*2.2.2   Calculating the steady-state weathering rate*
The reference weathering rate is calculated for each individual grid cell based on local steady-
state runoff $R_0$ (Figure 2a) and rock type composition. Following Amiotte Suchet and Probst
(1995), the local riverine fluxes of bicarbonate ions from carbonate $(f_{ca})$ and silicate $(f_{Si})$
weathering are computed as:

$$f_{ca} = R_0 \sum_i frac_i k_i \alpha_i \qquad (5)$$

$$f_{Si} = R_0 \sum_i frac_i k_i (1 - \alpha_i) \qquad (6)$$

where $frac_i$ is the fraction of rock type $i$ present in the grid cell, $k_i$ is the rock type specific
weathering rate multiplier, and $\alpha_i$ is the fraction of rock type given to weather as carbonate
rocks. The different rock types and their weathering parameters are shown in Table 1. The
weathering rate multipliers $(k_i)$ were derived from the data by Amiotte Suchet et al. (2003) and
the fractionation of rock types between carbonate and silicate rocks is adapted from the work of
Gibbs et al. (1999), following the interpretation of Colbourn et al. (2013).
The resulting steady-state carbonate and silicate weathering rates at pre-industrial (1800AD)
conditions are shown in Figure 2b. There is a noticeable concentration of CaCO$_3$ weathering in



areas of high runoff with bedrock composed predominantly of carbonate rocks (for example,
Southeast Asia), whereas $CaSiO_3$ weathering is spread more evenly across the world. It is
noteworthy that the Amazon basin features by far the highest runoff yet produces unremarkable
weathering rates (compared to other tropical areas) due to the prevalence of the weathering-
resistant shield rocks. The same observation can be used to explain the low weathering rates in
central Africa. This weathering distribution compares reasonably well with the $CO_2$
consumption distribution found by Amiotte Suchet and Probst (1995), but it doesn't reproduce
the large values at northern high latitudes (especially in northern Asia) that can be found using
the GEM-$CO_2$ model. The distribution of bicarbonate fluxes of Gibbs and Kump (1994) displays
a somewhat lower equator-to-pole gradient in weathering rates, and suggests an area of high
weathering in the southeast USA which is not reproduced with our model, mainly on account of
low runoff in the region. These discrepancies are likely due to precipitation bias in the UVic
ESCM. However, both models appear to agree with our finding that southeastern Asia is the
region with the highest regional weathering intensity. Globally, the 2-D weathering scheme
sends a DIC flux of 0.166 Pg C/y into the ocean, which is approximately 15% more than the 0-D
model output (0.145 Pg C/yr) (Meissner et al. 2012), and on par with previous estimations of
pre-industrial global weathering intensity.
*2.2.3   Modulation of weathering rate*
In transient model simulations, the carbonate and silicate weathering rate for each grid cell is
modulated by changes in local environmental conditions. They were made dependent on surface
air temperature, atmospheric carbon dioxide content, and runoff in a similar manner to previous
carbon cycling models. Following Lenton and Britton (2006), we have included the option of



replacing the dependency on CO2 concentration by vegetation productivity, which more directly
accounts for the impact of biological factors on weathering intensity.
Temperature is a known controller of weathering intensity as higher temperatures increase the
kinetic energy of molecules, facilitating the atomic encounters which lead to the chemical
dissociation of minerals.  Although it is impossible to derive a relationship between temperature
and weathering rates from first principles, laboratory and field studies have correlated the
concentration of bicarbonate ions in a solution to water temperature in order to develop an
empirical formulation.  For carbonate weathering, we used the results of Harmon et al. (1975),
who compared the groundwater temperature and bicarbonate ion concentration of several North
American watershed to come up with the following empirical relationship:

$$g_{ca}(SAT) = 1 + 0.049(SAT - SAT_0) \tag{7}$$

where $SAT$ and $SAT_0$ are the transient and steady-state surface air temperature, respectively.  For
silicate weathering, we used a version of the Arrhenius rate law of Brady (1991) which was
adapted into the RokGeM by Colbourn et al. (2013):

$$g_{si}(SAT) = e^{0.09(SAT - SAT_0)} \tag{8}$$

Here, the constant of 0.09 inside the exponential expression was obtained using an activation
energy of 63 kJ mol$^{-1}$ for silicate weathering and a global initial temperature of 288K (global
average pre-industrial temperature).  The activation energy is poorly constrained, but has been
shown to have little effect on the long-term consumption of atmospheric $CO_2$.
The productivity dependence of weathering serves to illustrate the biological and soil-
enhancement factors which control weathering intensity, with vegetation net primary
productivity a suitable proxy for biological activity in an area.  Lenton and Britton (2006)





introduced a simple linear dependence of weathering on productivity based on the steady-state
proportion of global productivity to global weathering flux:

$$g(NPP) = \frac{NPP}{NPP_0} \qquad (9)$$

where $NPP$ and $NPP_0$ are the transient and steady state net primary production, respectively,
taken explicitly from the output of the coupled land surface scheme MOSES-2 and vegetation
module TRIFFID. This formulation works reasonably well in 0-D models with globally-
summed values of productivity and weathering rate; however, some problems arise when trying
to use it in a spatially-explicit model due to its inherent assumption that productivity and
weathering intensity are directly related at steady state. A good example of this would be at the
continental margins of predominantly ice-covered continental landmasses (Greenland,
Antarctica) where some of the land may be ice-free, but too cold to support any vegetation.
However, the presence of nearby ice sheets generates a meltwater runoff flux which greatly
enhances weathering, in spite of the lack of vegetation. Therefore, any expansion of vegetation
in this area, however small (which is not unrealistic given the extreme warming scenarios
examined here), would result in an unreasonable increase in weathering. In order to rectify this
situation, we have introduced a modified version of equation 9 which calculates the increase in
local weathering rate when $NPP$ is greater than its steady-state value:

$$g(NPP) = \begin{cases} NPP/NPP_0, & NPP < NPP_0 \\ \left( 1 + \dfrac{(NPP - NPP_0)\big/NPP_{0,global}}{f_0\big/f_{0,global}} \right), & NPP \geq NPP_0 \end{cases} \qquad (10)$$

where $f_0 = f_{Ca} + f_{Si}$ and the "*global*" indices indicate the globally-summed value of that
variable. The right-hand term in the brackets is a compensation term, which modifies the





increase in weathering based on the relative contribution of the grid cell to the global
productivity compared with its contribution to the global weathering flux. This results in a
redistribution of NPP-induced changes in weathering without changing the globally-summed
increase in weathering intensity (this is true only in the absence of other controlling factors, such
as temperature). Note that the parameterization is unchanged from equation (9) whenever NPP
is lower than its initial value, mostly to avoid computing negative values of g(NPP). This has a
relatively benign impact on the global result, as the values calculated from equation (10) only
differ significantly from those of equation (9) when NPP is much greater than $NPP_0$.
As an alternative to productivity dependence, we also included the option to parameterize
weathering as a function of atmospheric $CO_2$ content, following the approach used in the
GEOCARB II model of Berner (1994):

$$g(CO_2) = \left( \frac{2\frac{pCO_2}{pCO_{2,0}}}{1 + \frac{pCO_2}{pCO_{2,0}}} \right)^{0.4} \tag{11}$$

where $pCO_2$ and $pCO_{2,0}$ are the transient and steady-state atmospheric concentration of $CO_2$,
respectively. This relationship has long been used to estimate the fertilizing effect of $CO_2$ on
land plants, and thus becomes here an indirect parameterization of the biological enhancement of
weathering. It can be used *in lieu* of equations 9 and 10 as a model option.
Runoff is the most widely used factor of weathering intensity as it constitutes a good proxy for
the strength of the water cycle in an area. One may consider the fact that high runoff
environments tend to be associated with intense weather activity (the rainforests, for example),
and also that stagnant waters quickly become saturated, thus limiting the efficiency of





weathering. Given that we already include runoff in the calculation of steady-state weathering,
the transient runoff dependency is a simple adjustment following *Berner* [1994]:

$$g_{Ca}(R) = \frac{R}{R_0} \tag{12}$$

$$g_{Si}(R) = \left(\frac{R}{R_0}\right)^{0.65} \tag{13}$$

where $R$ and $R_0$ are the transient and steady-state river runoff, respectively, which are also taken
explicitly from the output of the land surface scheme MOSES-2. The difference between the
formulations for carbonate and silicate weathering is an empirical correction based on the
assumption that bicarbonates from the weathering of silicate rocks are more diluted in rivers than
for carbonate weathering. The value of 0.65 in equation (13) was taken from Berner (1994);
although the value itself has a large margin of error, it has been shown to have only a modest
effect on the overall efficiency of the weathering feedback mechanism.
To summarize, we have developed a two-dimensional weathering scheme whereby the steady-
state values of carbonate and silicate weathering fluxes (see Sect. 2.2.2) are modulated by
changes in temperature, vegetation productivity (alternatively: atmospheric CO2 concentration),
and runoff. Thus the complete weathering parameterizations take the form:

$$F_{Ca} = f_{Ca} \cdot g_{Ca}(SAT) \cdot g(NPP) \cdot g_{Ca}(R) \tag{14}$$

$$F_{Si} = f_{Si} \cdot g_{Si}(SAT) \cdot g(NPP) \cdot g_{Si}(R) \tag{15}$$

Weathering is calculated in each individual land grid cell, and routed to the coastal ocean as
fluxes of alkalinity and dissolved inorganic carbon, explained in section 2.2.4.
*2.2.4   Effects of weathering on ocean biogeochemistry*



In the UVic ESCM, weathering does not have a direct impact on atmospheric or land surface
carbon; its effects are prescribed through the riverine exports of weathering products which are
sent to the ocean and modify its chemical composition.  The chemical weathering processes are
described by equations 1 (carbonate) and 2 (silicate), both resulting in a flux of two moles of
bicarbonate ions ($HCO_3^-$).  The flux of dissolved inorganic carbon ($F_{DIC}$) is counterbalanced by
the consumption of atmospheric carbon dioxide during the weathering reactions, leaving a net
DIC flux of one mole for carbonate weathering, and none for silicate weathering; a constant term
was also added to represent the contribution of volcanic outgassing to global carbon emissions
(which is not included in the UVic model):

$$F_{DIC} = F_{Ca} + F_{volc} \qquad (16)$$

Given that, in the absence of external forcings, the $CO_2$ consumption by silicate weathering is
meant to counter the intake of carbon from the geologic reservoir from volcanic eruptions, we set
the constant term $F_{volc}$ to equal the steady-state flux of silicate weathering ($F_{volc} = F_{Si,0}$).
Meanwhile, the net flux of alkalinity ($F_{ALK}$) remains equal to the flux of bicarbonate ions.  The
above discussion is summarized in the following set of equations, which describes the
partitioning of carbonate weathering and silicate weathering fluxes into dissolved inorganic
carbon and alkalinity fluxes, which are then globally summed and fed to the ocean
biogeochemistry module:

$$F_{DIC} = F_{Ca} + F_{Si,0} \qquad (17)$$

$$F_{ALK} = 2F_{Ca} + 2F_{Si} \qquad (18)$$

Note that our choice of $F_{volc}$ effectively equilibrates ocean biogeochemistry during equilibrium
runs ($F_{ALK,0} = 2F_{DIC,0}$).  The values calculated here represent net fluxes over the entire surficial
(atmosphere-land-ocean) reservoir, and in particular the simplification for the net flux of DIC is



based on the assumption that the consumption of $CO_2$ from the atmosphere is immediately
balanced by an equivalent uptake of carbon from the ocean. This would be true in general, given
that the timescale of the weathering negative feedback mechanism far exceeds that of
atmosphere-ocean mixing; however in the timescales considered here ($10^3$-$10^4$ yrs) there would
be some delay between the consumption of $CO_2$ from the atmosphere and the release of $CO_2$ in
the ocean following $CaCO_3$ burial. This delay would not significantly alter the impact of
weathering on atmospheric geochemistry, but could reduce by as much as 10% the rate at which
alkalinity increases in the ocean (Colbourn et al., 2013).
**2.3    Steady-state weathering and description of transient model simulations**
Pre-industrial steady-state weathering was obtained by integrating the model for over 20,000
years under year 1800 boundary conditions, using rock type dependency and distribution as
detailed in sections 3.1.1 and 3.1.2. Land-to-ocean weathering fluxes stabilized in less than $10^3$
years, on account of runoff being mostly computed from atmospheric output. However, the
fixing of deep ocean alkalinity and dissolved inorganic carbon (DIC) content would have
required as much as $10^5$ model years – an impossibly long simulation time given the level of
complexity of the UVic model. Hence we extracted the model steady state after $10^4$ years, but
kept the background steady-state run ongoing concomitantly with the transient model
simulations in order to correct the output of the latter based on changes in the former. Each
transient simulation was forced with the historical natural and anthropogenic carbon emission for
200 years; at year 2000 an additional 5000 Pg C were emitted over one year (unless otherwise
indicated), and carbon emissions were set to zero thereafter. All simulations were carried out for
a period of ten millennia, ending at year 12,000.



A total of eight model versions were integrated to year 12,000, which we classified into three
groups of experiments (see Table 1 for a description of all experiments). Group A (Section 3.1)
experiments investigate the impact of the intensity and span of the prescribed carbon emissions.
Simulation A0 is the basic emission scenario outlined in the above paragraph, and thus served as
the main control run for this paper. Simulation A1 is similar but used a more conservative
estimate of 1000 Pg C for future anthropogenic emissions. Finally, simulation A2 extended the
carbon emission total of 5000 Pg over a much longer period: emissions were increased linearly
until reaching double the current (year 2000) carbon emissions; the remaining carbon emissions
were then distributed evenly during the period from 2050 to 3000, then set to zero thereafter.
Although distributed over a longer period, the total carbon emissions remained unchanged from
our control run A0.
Group B experiments (Sect. 3.2) compare the various model representations of the biological
enhancement factor. In simulation B1, we replaced the NPP dependence term $g(NPP)$ in
equations 14 and 15 with $g(CO_2)$ from equation 11 on all grid cells. Although carbon dioxide
concentrations in the atmosphere are known to vary slightly across the surface of the Earth, in
the UVic model $pCO_2$ is a global term with no defined spatial variability. This effectively
removes the two-dimensionality of the biological feedback term, leaving temperature as the sole
spatially-explicit variable. Runoff does not vary much unless there are major changes in
hydrology or ice sheet distribution, neither of which were considered in our simulations. A
further simplification was made in simulation B2 by removing the biological enhancement factor
altogether and incorporating a parameterization that is only based on the temperature-dependent
part of our spatially-explicit scheme.





Group C experiments (Sect. 4.3) compare the relative importance of carbonate and silicate
weathering through their impacts on riverine fluxes of alkalinity and DIC. In simulation C1, we
eliminated the silicate weathering feedback ($F_{Si} = f_{Si}$), leaving only the carbonate weathering
part of the parameterization. Likewise, in simulation C2, the carbonate weathering feedback was
negated ($F_{Ca} = f_{Ca}$), isolating the impact of the silicate weathering feedback. Finally, in model
version C3 we eliminated both carbonate and silicate weathering feedbacks to maintain constant
weathering fluxes (at steady-state values) throughout the simulation. This last model version
effectively simulates the carbon sequestration potential of the oceans in the absence of the
weathering feedback mechanism.
For each of the model versions outlined above (with the exception of C1 and C2), an identical
setup was used with a zero-dimensional version of the weathering model whereby weathering
rates were calculated based on global, rather than local changes in the control parameters
(temperature, NPP, runoff); these 0-D model versions are identified in the figures using the "*"
notation (for example, "A0*" refers to the zero-dimensional version of simulation A0). The code
for these 0-D model versions was developed in an earlier study of terrestrial weathering changes
with the UVic model (Meissner et al., 2012).
**3      Results**
**3.1    Group A results**
The time series of $CO_2$ concentration in the atmosphere as well as weathering fluxes of carbon
and alkalinity are shown in Figure 3a for each of the pulse (group A) scenarios (solid lines), and
compared with results from similar scenarios using the 0-D version of the model (dotted lines).
For all simulations, the 2-D model was always more efficient in removing $CO_2$ from the



atmosphere than its 0-D counterpart. This can be partially explained by the initial global total
weathering being slightly higher in the 2-D model (see Section 2.2.2); however, this cannot
account for alkalinity weathering increasing nearly three times as much in the 2-D model as it
does in the 0-D version (interestingly, 0-D weathering rates seem to be slightly higher than 2-D
values for scenario A1). Instead, we propose that this is a natural consequence of using a two-
dimensional approach. Temperature, productivity, and runoff are closely related, as all three are
positively affected by the increase in atmospheric $CO_2$: temperature from the greenhouse effect,
NPP through the $CO_2$ fertilization of plants, and runoff as a result of both a temperature (hence
precipitation) increase and the $CO_2$-induced increase in plant water-use efficiency. Vegetation
productivity also reacts positively to increases in temperature in extratropical regions, although
this effect could be overcompensated for by an opposite reaction in tropical regions, where
temperatures exceed the threshold for optimized plant growth (Matthews et al., 2005; Matthews
et al., 2007). This means that areas which see a large increase in one variable will more often
than not see equally large increases in one or both of the other variables, further enhancing the
local increase in weathering rates. A zero-dimensional model would not be able to create this
effect because it uses globally summed or averaged variables. This is especially important with
regards to temperature, as the global average would be dominated by oceanic SAT changes,
which tend to be smaller than continental SAT changes.
Scenario A2, when compared to A0, suggests that the ability of the Earth system to recover from
anthropogenic emissions is essentially independent of the rate at which the emissions occur.
Atmospheric $CO_2$ concentrations recovered more slowly in the gradual emissions scenario at first
since weathering fluxes were not increased as much, but the gap between the two curves
gradually narrowed after A2 emissions ended at year 3000. We found a difference of 164 ppm in



atmospheric $CO_2$ concentration between A0 and A2 at year 12,000, which is nonetheless much
greater than the difference between the pulse and IPCC A2 simulations in Meissner et al. (2012),
in which a 5000 Pg C pulse was spread over 300 years. Given that the longer the carbon
emissions are spread out over time, the longer it takes for atmospheric $CO_2$ levels to catch up to a
pulse scenario, we can surmise that pulse scenarios would overestimate the ability of the
weathering feedback mechanism to remove carbon from the atmosphere in the next several
millennia, unless a way is found to mitigate anthropogenic emissions within the next century.
Time series of various carbon reservoirs for Group A scenarios are shown in Figure 3b. The
ocean reservoir content at year 12,000 was nearly identical for scenarios A0 and A2, indicating
that the ocean was even more indifferent to the rate of carbon emissions than the atmosphere.
Zero-dimensional model ocean carbon exceeded the 2-D output at around year 8500 for
scenarios A0 and A2, due to the fact that there was more carbon remaining in the atmosphere-
ocean system. It is interesting that the land and sediment carbon reservoirs behaved differently
from other reservoirs. The large carbon emissions (and associated temperature anomalies) in A0
and A2 appeared to have a counterproductive effect on land carbon content, which was not seen
under the more modest temperature increase of simulation A1. If anything, this results points out
the inability of the land reservoir to store any significant amount of excess carbon from the
atmosphere on millennial timescales. Finally, the sediment carbon curve also behaved somewhat
counterintuitively, as all three 2-D pulse scenarios produced a comparable increase of sediment
carbon content, regardless of emission rate or amount of carbon released. For the 0-D model
versions, total $CaCO_3$ buried mass increased more rapidly for scenario A1 despite lower amounts
of carbon emitted. These results arise because carbon burial depends on a delicate balance which
involves ocean temperature, alkalinity, and calcite concentration. Oceans in scenario A1





contained less $CO_2$ and $CaCO_3$, but were also cooler than in A0 and A2, which may explain why
the total accumulation of $CaCO_3$ sediments was comparable between all three scenarios.
Global changes in surface air temperature, vegetation NPP, and surface runoff are shown in
Figure 4 at various times during the 10,000 year simulation.  As seen in Figure 4a, the most
significant changes in temperature mostly occurred poleward of 60 degrees of latitude; however,
there were also increases in many tropical regions.  These results are to be expected given the
static nature of wind fields in the UVic model, which prevent a reorganization of atmospheric
circulation and thus trap the warm anomalies in the tropics.  Figure 4a (see also Figure 3c) also
reveals that the cooling effects of carbon sequestration were not felt until well after year 3000,
despite atmospheric $CO_2$ concentrations being decreased by nearly 1000 ppm between years
2000 and 3000; this is simply due to the thermal inertia of the ocean (Matthews and Caldeira,
2008).  By year 12,000, temperature anomalies across the globe became fairly uniform, with
every area averaging 1-2°C warmer than pre-industrial state.
Changes in vegetation net primary productivity are shown in Figure 4b.  Most of the world saw
an increase in vegetation activity from the direct effect of $CO_2$ fertilization, with the exception of
desert areas which remained the same (Africa, Asia) or become more arid (Australia).  Changes
in NPP also correlated well with changes in river runoff (Figure 4c); this is mainly a
consequence of the effect of increased $CO_2$ concentrations on plants, which optimises vegetation
water-use efficiency, leading to an increase in soil moisture and therefore runoff (Nugent and
Matthews, 2012; Cao et al., 2010).  The very large NPP increase in Indonesia around year 3000
was likely caused by the replacement of rainforest by the much more productive C4 grasses, and
further enhanced by a 1000-year legacy of high $CO_2$ fertilization.





The resulting impacts of these changes in temperature, vegetation productivity, and runoff on
$CaCO_3$ and $CaSiO_3$ weathering are shown in Figure 5. We found that changes in weathering
were more strongly correlated to changes in NPP, and to a lesser extent, runoff. The influence of
rock type distribution was also noticeable, especially on the carbonate/silicate weathering
partitioning, while temperature had an overall modest impact. Most areas saw a moderate to
high increase in both weathering types, with the exception of arid areas (deserts and ice caps)
which experienced a minor decrease in weathering. The most significant change occurred in
central western Asia (Kazakhstan), mirroring a moderate increase in vegetation productivity
coupled with a considerable temperature change during the third millennium CE. The anomaly
all but disappeared in later snapshots of the simulation, once global temperatures were no longer
warm enough to sustain such high levels of vegetation productivity. Indonesia also saw a large
increase in silicate weathering rates, on account of all three controlling parameters increasing by
a large margin in the area, coupled with a predominantly silicate-heavy lithology. In later stages
of the model simulation, weathering rate anomalies had mostly retreated to the tropical latitudes,
where productivity and runoff anomalies persisted the longest; elsewhere the increase in
weathering rates was reduced to below 10% of their value during the third millennium CE.
**3.2   Group B results**
The purpose of this group of experiments was to assess the importance of including a
parameterization for NPP (A0) rather than atmospheric $CO_2$ concentration (B1), or a weathering
scheme based exclusively on temperature and runoff feedbacks (B2). Model output for
atmospheric $CO_2$ concentration and weathering fluxes is shown in Figure 6a. The results
strongly suggest that using vegetation productivity rather than $CO_2$ as a proxy for biological
activity makes weathering fluxes much more sensitive to overall climate and environmental



changes. Weathering fluxes peaked around year 2200, and the increase for simulation A0 (using
NPP) was twice that for simulation B1 (using $CO_2$), and about three times larger than for
simulation B2 (with no biological effect); in other words, adding a NPP dependence tripled the
weathering increase compared to the case using a temperature dependence only. This is likely a
result of rapid vegetation expansion in the high latitudes and the appearance of warm-adapted
and more productive biomes in the temperate regions, which was taken into account in A0 but
not in the Group B model versions. As a result, the recovery time of atmospheric $CO_2$ levels was
much faster in A0, and vegetation productivity rapidly dropped below the levels of B1 and B2
(not shown). Interestingly, after year 7000 the weathering DIC flux in simulation A0 fell below
that of B1, indicating that from that point onward the parameterization in B1was more effective
in removing $CO_2$ from the atmosphere. This feature does not appear in the 0-D model results,
where DIC weathering fluxes always remained higher in the productivity-dependent model
version. As mentioned in section 3.1, it is possible that the increased effectiveness of the 2-D
weathering parameterization (compared to 0-D) is caused by the coincidence of large
temperature/runoff increases within areas that also see a large increase in vegetation NPP.
Figure 6b displays the time series of ocean and sediment carbon, as well as three parameters
which are used to analyze the evolution of calcite sedimentation in the model: $CaCO_3$
production, pore layer portion, and dissolution. In contrast to 0-D versions of the model, there
was a clear convergence of all three 2-D curves for ocean carbon content resulting from a
substantial drop in weathering rates during the latter half of the simulation period. The rate of
increase of $CaCO_3$ buried mass was slower in model versions with lower weathering rates; this
surprisingly differs from Fig 3b where there was not much difference between the three model
versions. However, the 0-D curves still displayed a significant lag behind their 2-D counterparts.





The pore layer portion remained unchanged during the first thousand model years after the
introduction of the 5000 Pg C pulse, thus mirroring the results of *Meissner et al.* [2012]. Higher
biological activity and carbonate concentration in the surface ocean due to warmer temperatures
was found to lead to a sharp increase in calcium carbonate formation and precipitation; this was
balanced by a rising of the carbon compensation depth (CCD) in the deep ocean fueled by the
rising acidity of the ocean, which increased the overall dissolution rate of calcite. As the more
immediate effects of the carbon emission pulse receded, oceans became cooler and calcite
formation weakened, while dissolution kept increasing for another 1000 years. This created an
unbalance in the $CaCO_3$ pore layer fraction which appears from year 4000 onward. Note that
even though dissolution rates in the deep ocean exceeded calcite production in the surface layer,
there was still an overall increase in $CaCO_3$ buried mass due to the enormous increase in oceanic
carbon content.
Figure 7 displays the spatial distribution of $CaCO_3$ and $CaSiO_3$ weathering changes at various
points of the simulation timeline for model version B1. Several areas of higher weathering from
Figure 5 were completely absent (central Eurasia), and some others were greatly reduced
(tropical Africa). These are the most important examples of how the vegetation productivity
parameterization can greatly enhance carbonate weathering locally, and silicate weathering
worldwide (see Figure 4b). Weathering rates were generally higher in A0 throughout the
simulation, but it should be noted that in some areas in the final snapshot (year 12,000 CE)
weathering in A0 fell below that of B1. Since the only difference between B1 and B2 is the
presence of (globally averaged) atmospheric $CO_2$ concentration as a factor, the equivalent figure
for model version B2 (not shown) would have been extremely similar to B1.
**3.3    Group C results**



The purpose of this group of experiments was to isolate and compare the individual contributions
of carbonate (C1) and silicate (C2) weathering to the global feedback mechanism, and to
compare them with a scenario where this negative feedback does not exist (C3). The time series
of atmospheric $CO_2$ concentrations and DIC/alkalinity weathering are shown in Figure 8a. After
500 years of roughly similar behavior, the curves diverged into three distinct narratives. By year
12,000, about 33% of the emitted carbon was still in the atmosphere for model version C3
(constant weathering), whereas about 20% of the carbon remained for the C1 (change in
carbonate weathering only), and 10% for C2 (change in silicate weathering only) and A0 (control
run). This is due to the immediate effect of carbonate weathering, which increase alkalinity
content in the ocean faster than the rate at which the precipitation of calcium carbonate increases.
Over timescales of $10^5$ years or more, we would expect the C3 curve to catch up to C1 as
increased calcite burial releases carbon dioxide back to the ocean, negating the carbon removal at
the surface; this outcome is verified in the million-year simulations of Colbourn et al. (2013) but
impossible to replicate here due to the time scales involved. The C2 model version yielded very
similar results to A0, which included the impacts of both carbonate and silicate weathering. The
difference between the two was greater initially, as the additional alkalinity provided by
carbonate weathering further enhances the oceanic uptake of $CO_2$ from the atmosphere, but the
gap gradually narrowed as the medium-term impacts of carbonate weathering faded away.
Again this outcome is verified over geologic timescales by Colbourn et al. (2013), with both the
A0 and C2 equivalents returning the Earth system to pre-industrial levels. There was no change
in DIC weathering output from C2 since silicate weathering in this model does not increase the
DIC flux to the ocean. Alkalinity flux from C1 exceeded that of other model versions towards



the end of the simulation period as the slower carbon removal resulted in much warmer surface
conditions compared to other model versions.
The evolution of ocean sedimentation is presented in Figure 8b using five model variables: ocean
carbon, calcite buried mass, upward/downward flux of calcite, and pore layer portion. Ocean
carbon levels remained similar between C1 and C2 during the first two thousand years, after
which C1 overtook C2 and eventually A0, despite resulting in the least amount of carbon
removal of the three model versions. This is because C1 would send as much DIC into the ocean
as alkalinity, which is counterproductive to atmospheric carbon removal. Inversely, C2 removed
almost as much carbon as A0 while adding 1000 Pg C less into the ocean, testifying towards the
efficiency of silicate weathering in removing carbon from the atmosphere compared to carbonate
weathering. Calcite sedimentation followed a very similar evolution to the experiments in Group
B (section 3.2), with the calcite pore layer portion remaining unchanged for 1000 years until the
upward flux of calcite (dissolution) became larger than the downward flux of calcite
(production/precipitation). Here it becomes clear that rock weathering, and in particular, silicate
weathering, is crucial in maintaining the stability of the pore layer fraction in the long term.
Model version C3, where weathering rates remain constant, produced a much sharper increase in
calcite dissolution compared to A0, where both weathering types respond to changes in climate,
and the burial of $CaCO_3$ in sediments occurred much faster in A0 than in C3. Additionally, pore
layer portion was better maintained by silicate weathering (C2) than carbonate weathering (C1).
These results suggest that the alkalinity flux supplied by silicate weathering is necessary not only
for decreasing the oceanic buffer factor (i.e., the concentration of carbonic acid and carbonate in
ocean surface water) and allowing the uptake of more $CO_2$ from the atmosphere, but also to



maintain a better balance of the oceanic sedimentary pore layer by mitigating the increase in
calcite dissolution in the deep compared to the production rate in the surface layer.
**4      Discussion**
The weathering scheme introduced here is subject to some caveats relating to the formulation
itself, as well as the limitations inherent in the UVic model.  Colbourn et al.  (2013) carefully
discuss the potential misrepresentation of temperature as a factor, especially when other
parameters such as vegetation productivity are also taken into account.  In particular, it is
possible that the temperature dependency for carbonate weathering (equation 7), which was
developed empirically from correlating groundwater $CaCO_3$ concentration with water
temperature in various river catchment basins, also captures the coincident changes in vegetation
productivity and river runoff, hence making the other factors redundant to a certain extent.
Whether this would introduce a significant error to the modeling is questionable, as temperature
on its own was shown to have at most a moderate impact on overall changes in weathering rates
(see Sect. 3.1).
The validity of the other two parameterizations – NPP and runoff – is difficult to assess as the
formulations are based on the arbitrary assumption that weathering rates vary monotonically and
linearly with changes in the two parameters.  In the case of productivity dependence, for
example, the parameterization is meant to represent the physical impacts of root expansion, and
the chemical impacts of soil kinetics, on the breakdown of rock into minerals and their eventual
dissociation by carbonic acid.  Thus an ideal productivity scheme would account for the impacts
of various plant types on each of the lithologies in terms of areal coverage, root expansion, and
other relevant quantities.  Moreover, it should be noted that both the NPP and runoff schemes in



our model rely heavily on the ratio between initial weathering and initial NPP/runoff, meaning
that a change in a parameter in an initially low-activity region (such as colder climates) may have
a disproportionately higher impact on weathering rates compared to changes in tropical areas.  It
is possible that the introduction of vegetation in a previously nonvegetated area would introduce
a stress likely to drastically increase rock erosion, but this is an effect that would be better
represented by directly parameterizing the new plant type as a stress on the underlying lithology.
Therefore, a better parameterization may be one based on the absolute value of NPP/runoff
(using for example, a non-linear empirical function linking weathering rate and net primary
productivity) rather than the ratio of the current value to the initial value.  The development of
such a relationship, however, would require a more in-depth investigation of the role of plants,
and biotic activity in general, on the physical and chemical erosion of rocks.
Another source of uncertainty in our results lies in the UVic model itself.  While very well suited
to simulate long-term impacts of carbon emissions and increased weathering rates on ocean
biogeochemistry, on a shorter time scale ($10^2$-$10^3$ years) the lack of advanced atmospheric
dynamics prevents the model from adapting to the extreme warming brought on by carbon
emissions in a manner consistent with our understanding of global climate.  Under extreme
warming there is a poleward shift of the tropical and subtropical cells and consequent changes in
precipitation patterns, leading to a potential overestimate of atmospheric temperature and
moisture content changes over tropical regions (see Sect. 3.1).  This effect is important mostly
between years 2000-3000 CE, and fades away as the brunt of the climate and biogeochemical
changes are shifted to the oceans.  The model's simplified precipitation scheme also likely
affects its ability to simulate runoff changes, which are central to both the initiation and
modulation of weathering rates.



Terrestrial rock weathering is a complex mechanism with many variables worth considering,
many of which have a high degree of interdependence. In the scheme introduced in this paper
for the UVic model, we considered the impacts of temperature, productivity and runoff (all
parameters previously examined in zero-dimensional weathering models), along with lithological
distribution to drive spatial variability. However, many other factors which affect weathering
rates were unaccounted for that could also be relevant in the context of a spatially explicit
weathering scheme. Perhaps the most meaningful of all is the consideration of sea level change.
It is highly likely that the extreme warming caused by anthropogenic emissions would result in a
significant melting of the Greenland and West Antarctic ice sheets (Clark et al., 2016), not only
disrupting the freshwater balance in the polar oceans, but also greatly contributing to a global
rise in sea levels along with the thermal expansion of seawater. Many of the low-elevation
continental shelves threatened by sea level rise are situated in weathering active, tropical regions,
and therefore the interruption of terrestrial weathering due to the flooding of these areas could
substantially reduce the global weathering output, thus weakening the response to global
warming. Note that the extensive warming could also bring about a decrease in ice sheet area,
especially in Greenland, which would open up some potentially very active weathering regions
(Kump and Alley, 1994). However, the extent of this areal reduction of ice sheet cover over a
few thousand years is likely to be overwhelmingly compensated by the area of land flooded by
sea level rise.
Another factor of some relevance is the interaction with land biogeochemistry. There has been
an extensive discussion in recent years on the role of mid- to high-latitude peatlands in the
context of a rapidly warming Earth, especially with regards to the decarbonation of these
ecosystems and subsequent release of greenhouse gases in the atmosphere (mostly methane) that



could greatly amplify global warming. While the release of methane by itself does not directly
affect terrestrial weathering, there are a variety of soil processes within peatlands which are
triggered or amplified by warming and which would have a significant local effect on the
chemical dissociation of rocks.
There are many other factors which would be worth investigating. For example, a distinction
between physical and chemical weathering would allow the inclusion of factors such as altitude,
as wind and relief/slope play a major role in physical weathering. The impact of ground frost at
higher latitudes also leads to erosion, and could increase weathering rates in colder climates.
Finally, one cannot ignore anthropogenic impacts, in particular the spread of modern agriculture,
in which crop yields are often boosted using mineral fertilisers and other chemicals, which mix
in with the soil water and accelerate the erosion of the bedrock. Other features of the
Anthropocene worth mentioning include acid rain and land use change, all of which need to be
taken into consideration in order to better represent the modern dynamics of global
biogeochemistry. Unfortunately, it is unlikely that most of these factors can be properly
integrated in current low- and intermediate-complexity climate models, on account of their
requiring a spatial resolution much finer than what most EMICs can offer. For example, the
UVic model's 1.8°×3.6° resolution cannot resolve physical mechanisms which occur over a
single-kilometer spatial scale.
**5    Conclusions**
A spatially-explicit weathering scheme has been developed and integrated into the University of
Victoria Earth System Climate Model (UVic ESCM). The model was constructed in such a way
that weathering rates at a certain point are computed based on the difference in temperature,





vegetation net primary productivity, and runoff, between that point and pre-determined initial
conditions. In our model, those initial conditions were based on pre-industrial runoff and
lithology (Amiotte-Suchet et al., 2003), which provides the basis for the two-dimensionality of
the model.
The model was tested with scenarios of future climate change, using (in most cases) a pulse of
5000 Pg C at year 2000 to simulate climate system recovery from anthropogenic emissions and
the role of global weathering during the following 10000 years. Overall, the model results
suggested that weathering has a negligible effect on atmospheric $CO_2$ and ocean biogeochemistry
on short timescales, but its impact becomes more discernible as we progress to multimillennial
timescales. We also found that climate system recovery from carbon emissions was much faster
using a two-dimensional model rather than the zero-dimensional model versions used in previous
work. Among the various climate factors used in the model, we found primary productivity to be
by far the most important, producing an increase in global weathering far higher than a model
version using atmospheric $CO_2$ levels to represent biotic activity, or one where only temperature
and runoff changes were considered. This highlights the need for further research to determine
whether this effect of biotic activity on physical and chemical weathering is in fact an important
real-world process that is independent of temperature and/or runoff change. Lithology itself was
also found to be very important, often meaning the difference between a weathering-active and
high- and low-weathering region. In terms of global totals, carbonate weathering was found to
be more prominent than silicate weathering, mostly on account of weathering-vulnerable rocks
being mostly carbonate-weathered. However, our results clearly emphasized that silicate
weathering is the only process of the two which has the capacity to fully restore the climate





system to pre-industrial levels (on timescales of $10^5$ years), thus confirming the findings of
Colbourn et al. (2013).
This work has established the importance of using a spatially-explicit weathering scheme to
better represent long-term changes in carbon biogeochemistry. Our approach, although crude,
has demonstrated that weathering can be integrated on the grid-cell level and still produce
reasonable results. This study did not take into account the more subtle aspects of spatial
variability, such as the impacts of ice sheets, sea level changes, and local factors such as soil
activity and topography. These are therefore important processes to include in further analyses
of the effect of deglacial weathering changes on ocean biogeochemistry and climate change.
**Acknowledgements**
We thank P. Amiotte-Suchet for providing the dataset for the worldwide rock type distribution
for the GEM-$CO_2$ model, as well as K. Meissner for providing the code for 0-D model versions,
A. Mucci for helping us better understand carbonate chemistry, and G. Colbourn for his
assistance in understanding the RokGeM model. The support of NSERC Discovery Grants
awarded to LAM and HDM is also gratefully acknowledged.





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



**Table captions**
**Table 1.** Rock type constants used in equations 5 and 6. Here, $k$ represents the weathering rate
multiplier, with a higher multiplier signifying a higher concentration of weathering products per
unit of runoff (or alternatively, a lower resistance to weathering agents); $\alpha$ denotes the fraction of
the given rock type to weather as carbonate rocks. A value of zero means that the rock type
consists of silicate minerals only.
**Table 2.** Description of each experiment carried out in this study. The emission total is the total
amount of carbon emitted by the pulse starting at year 2000, while the emission period represents
the time span of the pulse; the emission total is divided equally among the number of time steps
during the emission period. For the biological factor, "NPP" refers to equation 10, "Atm. $CO_2$"
to equation 11, and "None" signifies that this part of the weathering scheme is ignored. Finally,
when the $CaCO_3$ switch is OFF, the amount of carbonate weathering produced by the model is
set to its pre-industrial value for the duration of the simulation; similarly for when the $CaSiO_3$
switch is OFF.



**Figure captions**
**Figure 1.** Distribution of the six major rock types for the present day. Presented here are **(a)** the
source data from GEM-CO$_2$ [*Amiotte Suchet et al.*, 2003] in 1°×1° resolution; **(b)** its adaptation
to the UVic model in 3.6°×1.8° resolution, displaying only the dominant lithology in each grid
cell; and **(c)** the interpolated rock type fraction in each grid cell. For the latter, the data is shown
ranging from 0 (white) to 1 (full color).
**Figure 2.** Pre-industrial (year 1800 CE) setup for our weathering scheme. **(a)** Model simulated
annual mean river runoff, which is combined with rock type fractions (Figures 1 (b) and (c))
using equations 5 and 6 to produce **(b)** the carbonate and silicate weathering fluxes at pre-
industrial steady-state. Note the non-linear color scales, used here to better display values
outside of tropical regions.
**Figure 3.** Time series of simulated changes in various globally-averaged or summed model
outputs for Group A scenarios, compared with pre-industrial steady-state values. These
scenarios include **A0** (high-amplitude pulse), **A1** (low-amplitude pulse), **A2** (gradual emissions),
and their zero-dimensional counterparts (indicated by the "*" symbol). Shown here are (from
top to bottom): **(a)** atmospheric CO$_2$ concentrations, and weathering fluxes of DIC and alkalinity;
**(b)** global carbon budgets for atmospheric, ocean, land, and sediment reservoirs; and **(c)** surface
air temperature, net primary productivity, and oceanic concentrations of DIC and alkalinity.
Note the different scales along the time axis (separated by vertical dashed black lines). The



curves shown here represent experiments A0 (red), A1 (blue), and A2 (green), with dashed
colored lines representing the zero-dimensional equivalent model version.
**Figure 4.**  Spatial distribution of changes in **(a)** surface air temperature; **(b)** vegetation net
primary productivity; and **(c)** river runoff for experiment A0 from pre-industrial (year 1800 CE)
state to years 2100, 3000, 6000, and 12000 CE.  Non-linear color scales are used in panels (b)
and (c) to better display the results for the later stages of the model simulation.
**Figure 5.**    Spatial distribution of changes in carbonate ($CaCO_3$) and silicate ($CaSiO_3$)
weathering for experiment A0 from pre-industrial (year 1800 CE) steady-state to years 2100,
3000, 6000, and 12000 CE.  Note the non-linear color scale, used to better display values during
the later stages of the model simulation.
**Figure 6.**  Time series of simulated changes in various model outputs for Group B scenarios,
compared with pre-industrial steady-state values.  These scenarios include **A0** (dependence on
temperature, NPP, and runoff), **B1** (dependence on temperature, atmospheric $CO_2$, and runoff),
**B2** (dependence on temperature and runoff only), as well as their zero-dimensional counterparts
(indicated by the "*" symbol).  Shown here are (from top to bottom): **(a)** atmospheric $CO_2$
concentrations, and weathering fluxes of DIC and alkalinity; and **(b)** oceanic carbon budget,
sediment carbon budget, downward flux of calcite into sediments, calcite pore layer portion, and
dissolution of calcite in sediments.  Note the different scales along the time axis (separated by





vertical dashed black lines). The curves shown here represent experiments A0 (red), B1 (blue),
and B2 (green), with dashed colored lines representing the zero-dimensional equivalent model
version.
**Figure 7.** Spatial distribution of changes in carbonate ($CaCO_3$) and silicate ($CaSiO_3$) weathering
changes for experiment B1 between pre-industrial steady-state (year 1800 CE) and years 2100,
3000, 6000, and 12000 CE. Note the non-linear color scale, used to better display values during
the later stages of the model simulation.
**Figure 8.** Time series of simulated changes in various model outputs for Group C scenarios,
compared with pre-industrial steady-state values. These scenarios include **A0** (both weathering
types active), **C1** (carbonate weathering only), **C2** (silicate weathering only), **C3** (no
weathering), as well as the zero-dimensional counterparts to C0 and C3 (indicated by the "*"
symbol). Shown here are (from top to bottom): **(a)** atmospheric $CO_2$ concentrations, and
weathering fluxes of DIC and alkalinity; and **(b)** oceanic carbon budget, sediment carbon budget,
downward flux of calcite into sediments, calcite pore layer portion, and dissolution of calcite in
sediments. Note the different scales along the time axis (separated by vertical dashed black
lines). The curves shown here represent experiments A0 (red), C1 (blue), C2 (green), and C3
(black), with dashed colored lines representing the zero-dimensional equivalent model version
(when available).





1    **Table 1**

| Lithology | $k$ | $\alpha$ |
|---|---|---|
| Carbonate rocks | 1.586 | 0.93 |
| Shales | 0.627 | 0.39 |
| Sands and sandstones | 0.152 | 0.48 |
| Basalts | 0.479 | 0 |
| Shield rocks | 0.095 | 0 |
| Acid volcanic rocks | 0.222 | 0 |



1 **Table 2**

| Group | Experiment Name | Emission total (Pg C) | Emission period (years CE) | Biological parameter | CaCO$_3$ weathering switch | CaSiO$_3$ weathering switch |
|---|---|---|---|---|---|---|
| **A** | A0 | 5000 | 2000-2001 | NPP | ON | ON |
| | A1 | **1000** | 2000-2001 | NPP | ON | ON |
| | A2 | 5000 | **2000-3000** | NPP | ON | ON |
| **B** | B1 | 5000 | 2000-2001 | **Atm. CO$_2$** | ON | ON |
| | B2 | 5000 | 2000-2001 | **None** | ON | ON |
| **C** | C1 | 5000 | 2000-2001 | NPP | ON | **OFF** |
| | C2 | 5000 | 2000-2001 | NPP | **OFF** | ON |
| | C3 | 5000 | 2000-2001 | NPP | **OFF** | **OFF** |

3  Note: The "*" notation refers to zero-dimensional versions of the model using otherwise
4  identical experimental parameters.



1    **Figure 1a**

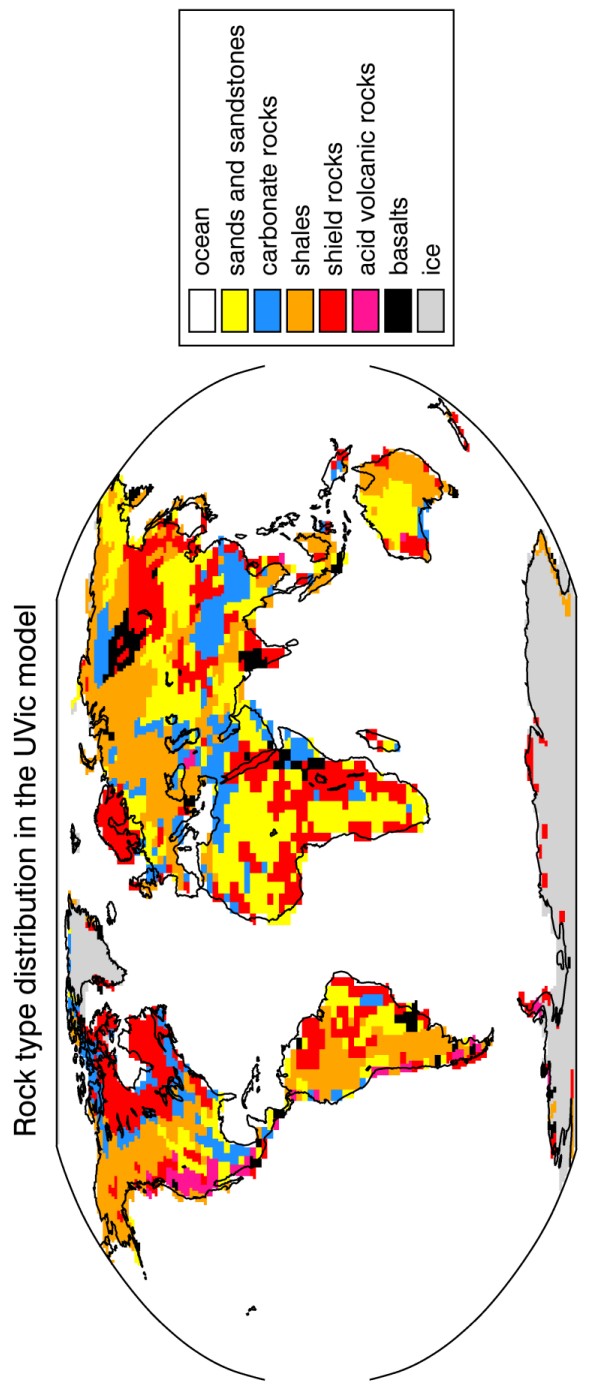



1 **Figure 1b**

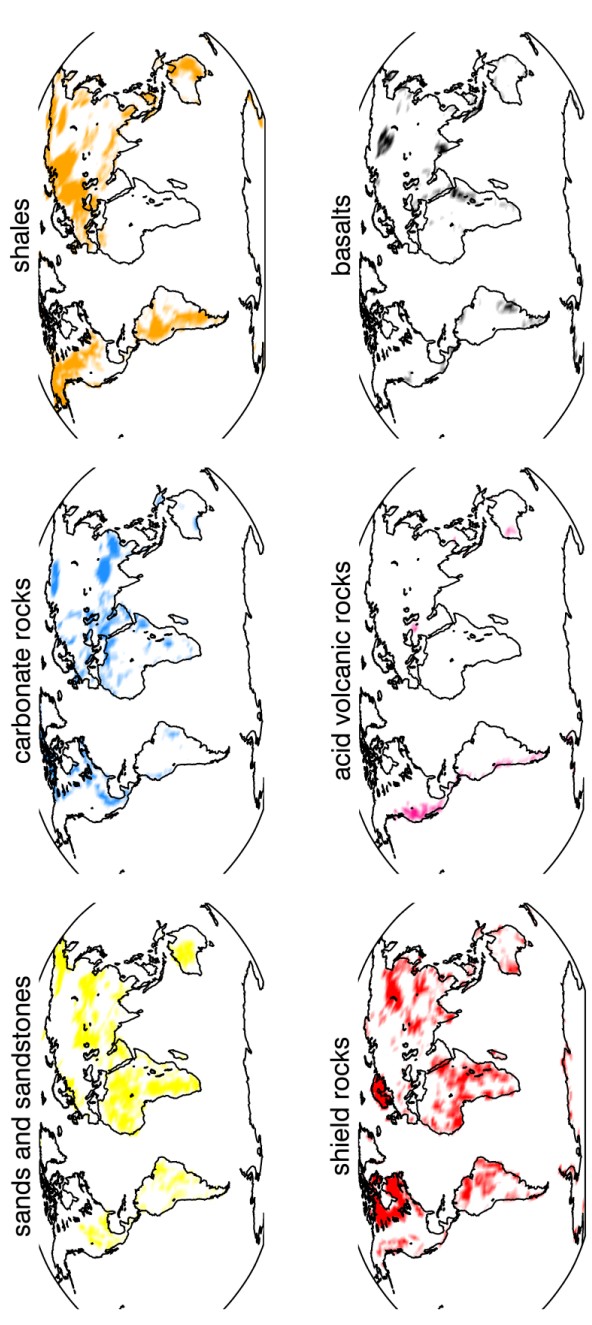



1   **Figure 2a**

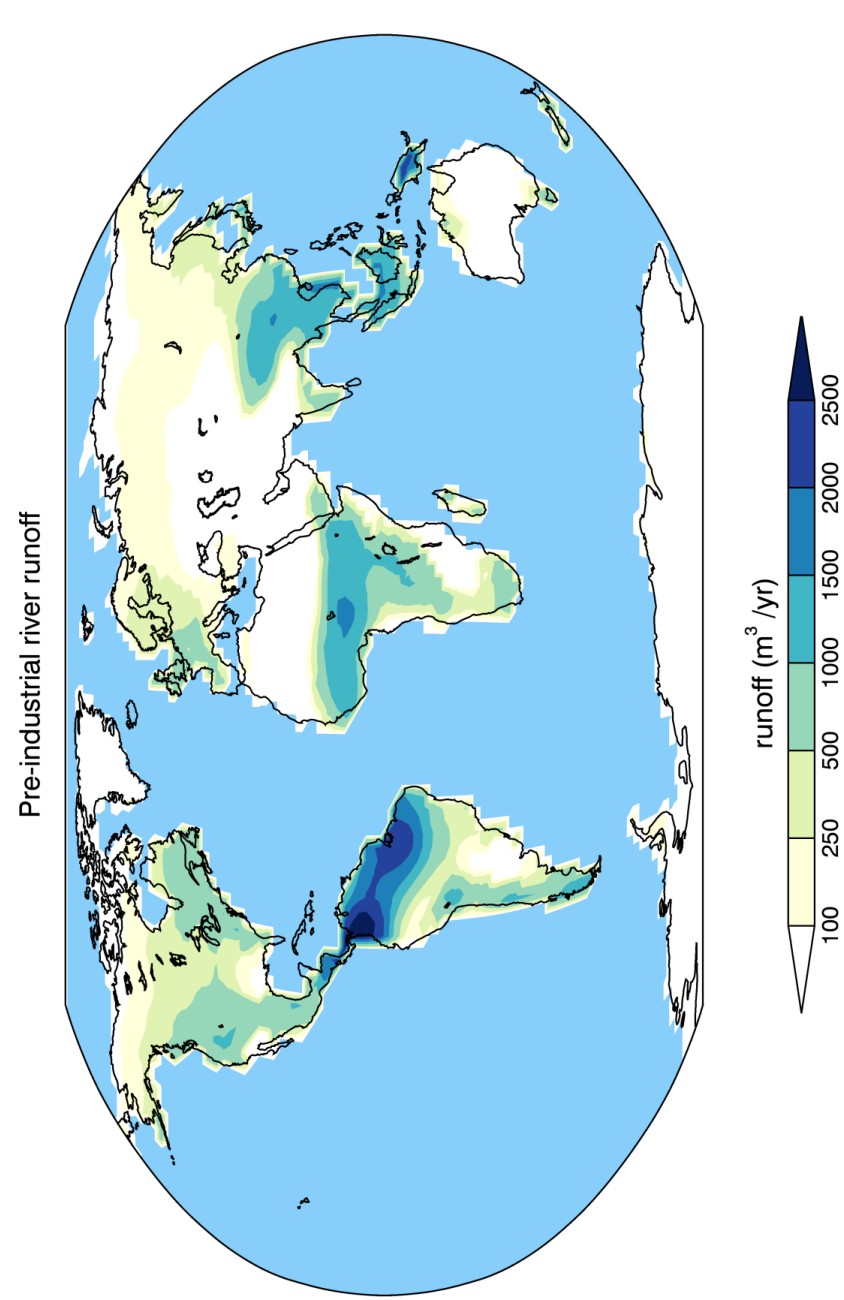



1    **Figure 2b**

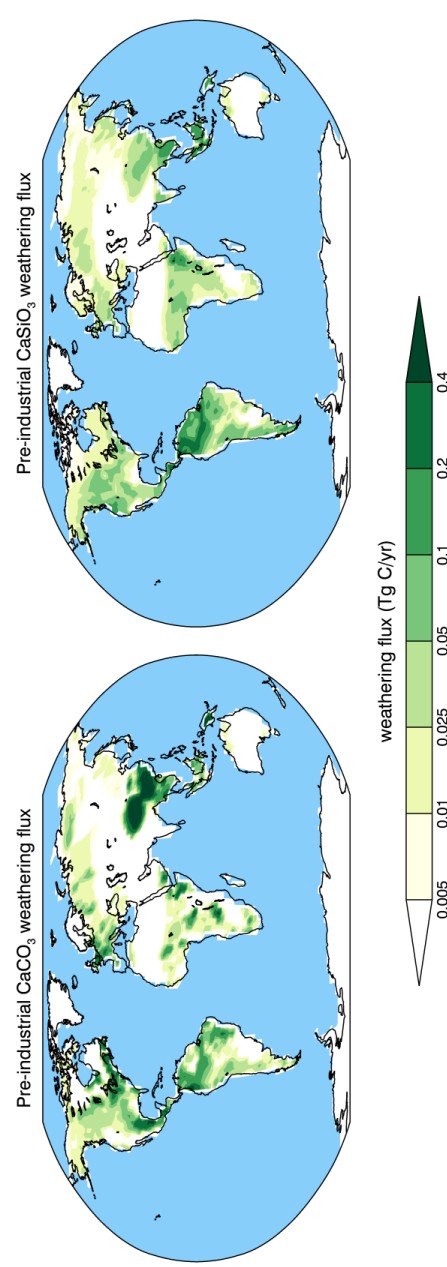



1  **Figure 3a**

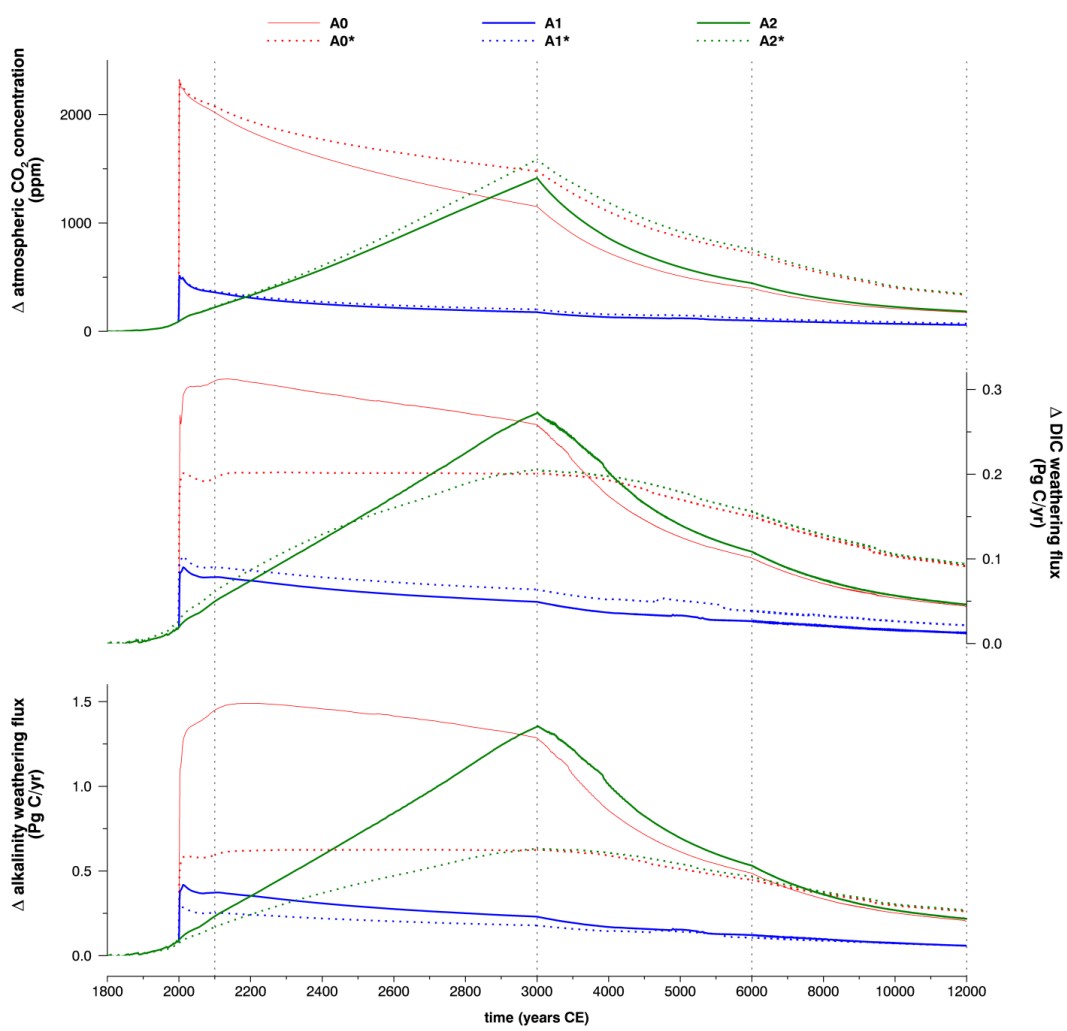





1    **Figure 3b**



1    **Figure 3c**



1  **Figure 4a**

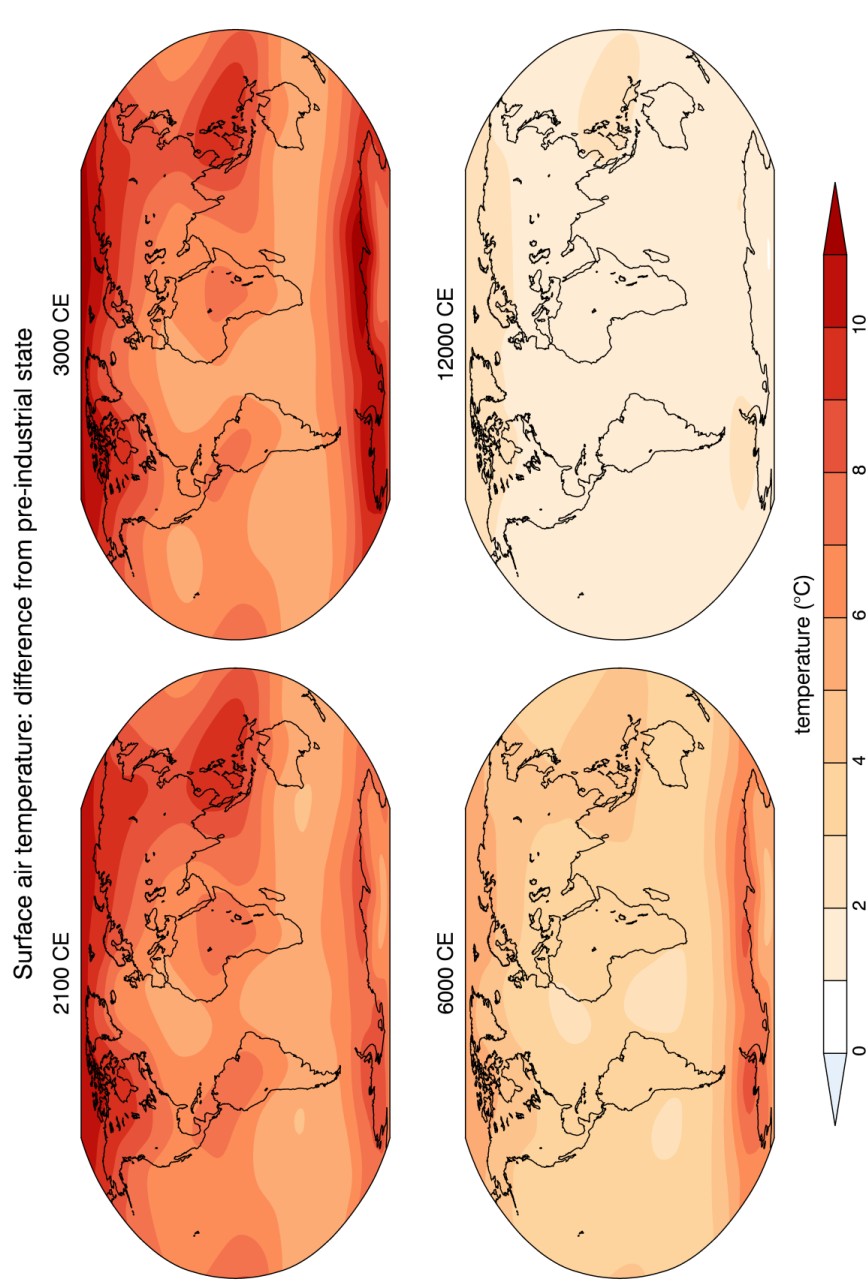



1    **Figure 4b**

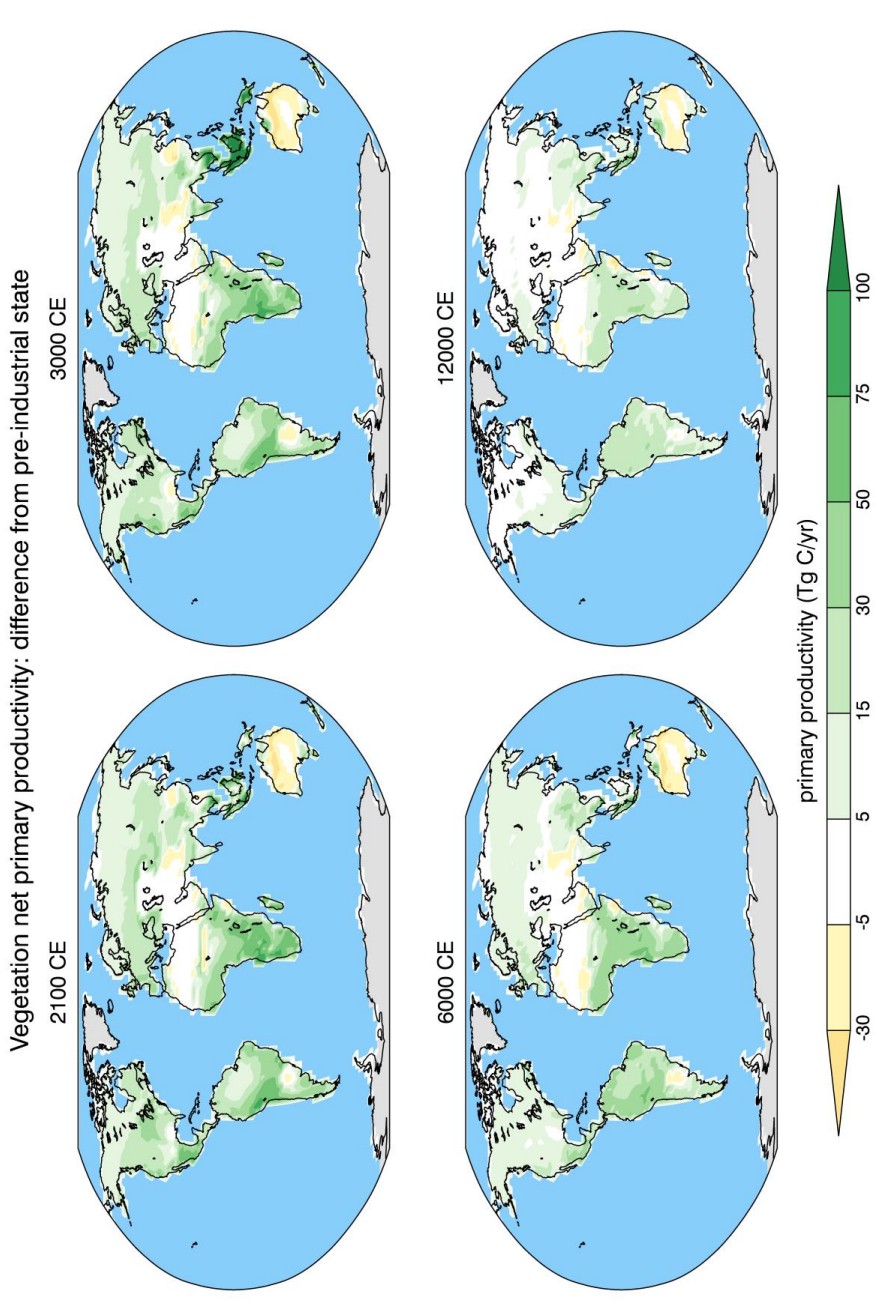




1   **Figure 4c**

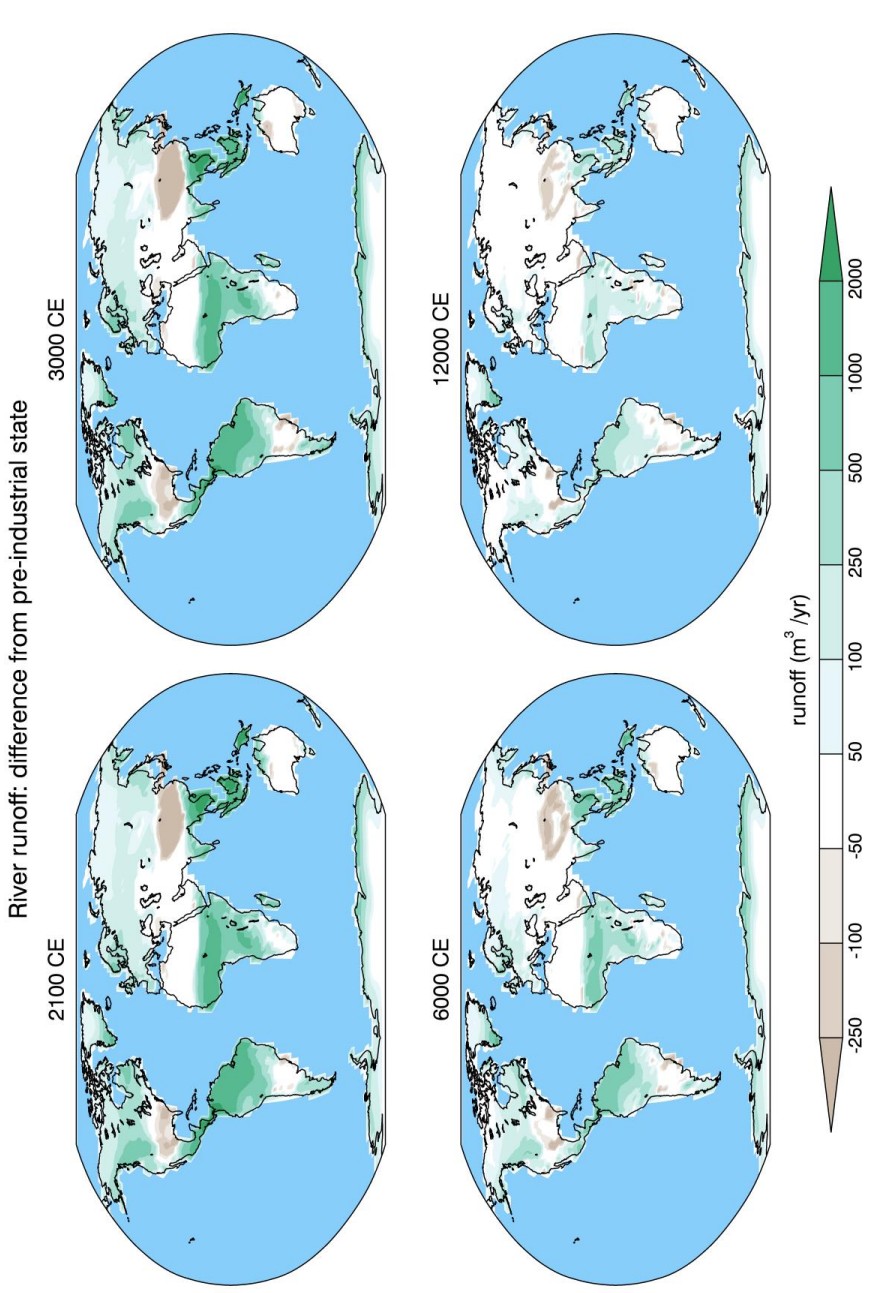




1 **Figure 5**

![weathering flux difference from pre-industrial state figure]



1    **Figure 6a**

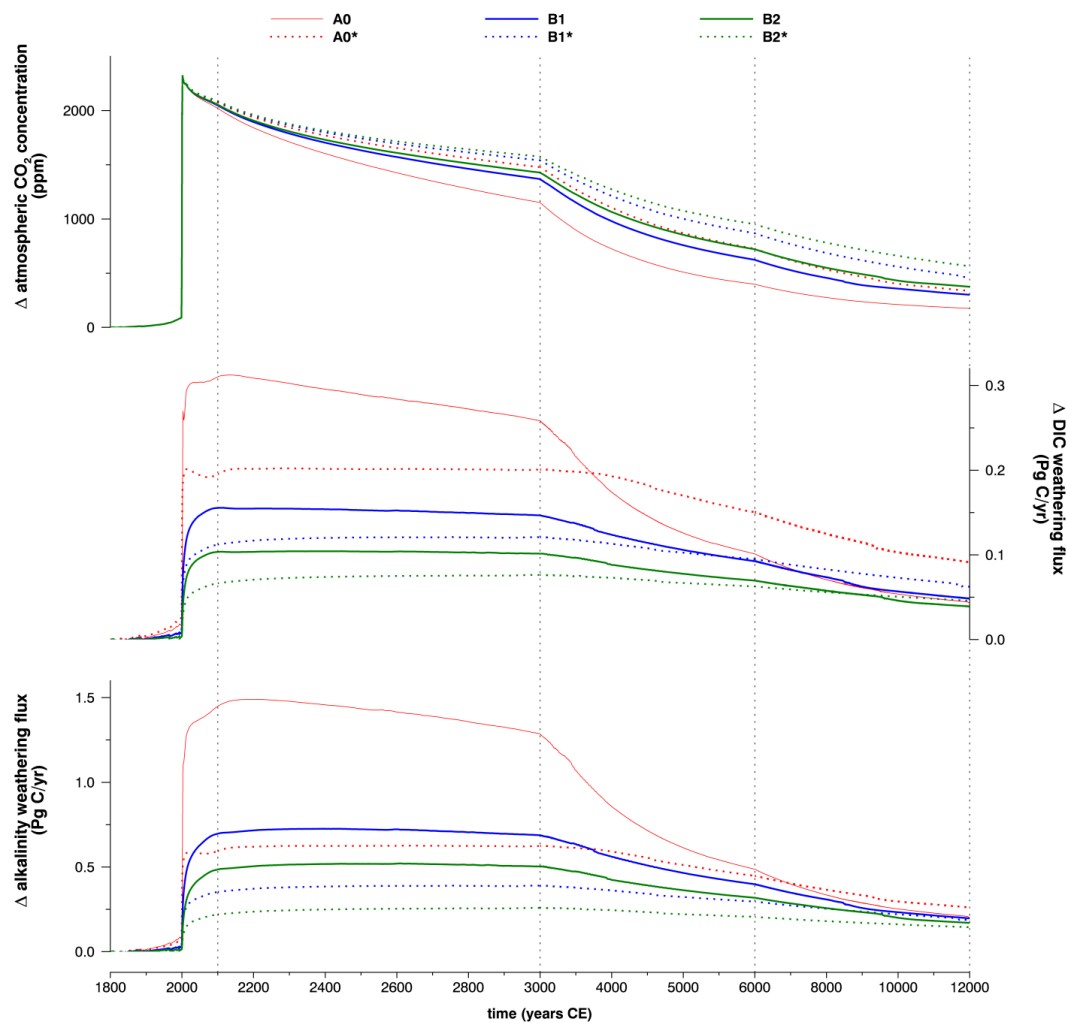



1    **Figure 6b**





1  **Figure 7**

weathering flux difference from pre-industrial state

_CaCO_3 weathering flux_ _CaSiO_3 weathering flux_





1 **Figure 8a**

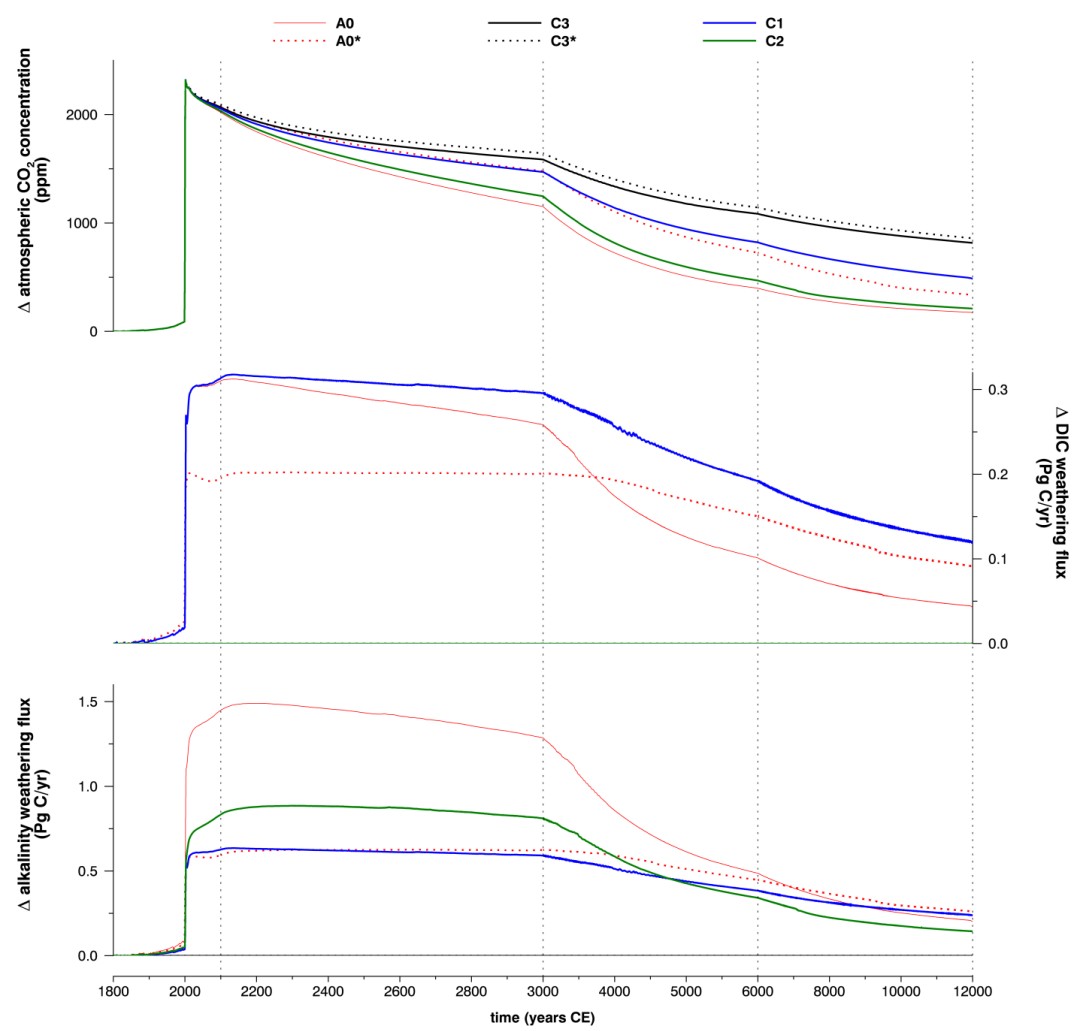



1    **Figure 8b**

