# Peer review of "The importance of terrestrial weathering changes in multimillennial recovery of the global carbon cycle: a two- 3 dimensional perspective"

_Earth System Dynamics, 2016_

## Referee Comment (RC1) · Anonymous Referee #1 · 29 Jan 2017

Brault et al. built spatially explicit parameterization schemes of continental weathering into the UVic Earth system model of intermediate complexity. Changes in weathering rate are parameterized based on rock types, changes in surface temperature, terrestrial primary production, river runoff, and atmospheric CO2. Then the UVic model is used to project long-term changes in the global carbon cycle under assumed anthropogenic CO2 emission scenarios. Simulation results using spatially explicit weathering schemes are compared with those parameterizing weathering flux using global mean variables. In general, it is found that the terrestrial weathering has a negligible effect on the ocean biogeochemistry and climate change on the timescales from decades to

centuries, but become much more important on multimillennial timescales. This finding is qualitatively in line with earlier modeling studies that incorporates a 0D weathering component.

The novelty of this study is the representation of terrestrial weathering on each model grid cell, which is an improvement over earlier studies using 0D weathering scheme. Given the large uncertainty of the weathering parameterization schemes (The authors did a nice job in discussing those uncertainties, though) and the model-simulated changes in relevant variables over the timescale of many thousands of years, it is difficult to make an assessment of how reliable the quantitative results are. Nevertheless, this work is one of the few Earth system modeling studies that provides a long-term projection of the global carbon cycle and climate with a spatially explicit weathering component. This manuscript is well written. I recommend its publication with a few minor comments.

Specific comments:

The six-page introduction is a comprehensive review of the rock weathering processes, in particular the modeling studies of weathering effect on the carbon cycle. This kind of introduction is useful for broad readers to understand the weathering effect. However, this type of introduction might be too lengthy to fit the journal. The authors may need to condense it.

Page 3, lines 10-11: "However, there have been but very few quantitative assessments of its impacts on carbon cycling and ocean biogeochemistry.."

"very few quantitative assessment" is not an objective statement. There are actually quite a number of existing modeling studies, as cited in the following section, on the effect of rock weathering on the carbon cycle.

Page 11, lines 4-5 "The UVic ESCM also includes a fully coupled global carbon cycle, which consists of inorganic carbon chemistry and land-surface exchanges of $CO_2$

(Ewen et al., 2004)"

"land-surface exchanges of CO2" should be "air-sea exchange of CO2"

Page 15, lines 16-17: "and on par with previous estimations of pre-industrial global weathering intensity."

Some references should be given here for the data-based estimate of pre-industrial global weathering intensity.

Page 19, lines 14-15: " Weathering is calculated in each individual land grid cell, and routed to the coastal ocean as fluxes of alkalinity and dissolved inorganic carbon"

It would be helpful to elaborate a bit more of how the river routing is done in the UVic model.

Page 21, " Pre-industrial steady-state weathering was obtained by integrating the model for over 20,000 years under year 1800 boundary conditions ... However, the fixing of deep ocean alkalinity and dissolved inorganic carbon (DIC) content would have required as much as 105 model years – an impossibly long simulation time given the level of complexity of the UVic model."

It would be useful to show something like time series of ocean-mean DIC and alkalinity to see how far the system is away from equilibrium after 20,000 years of spinup.

Page 25, lines 22-23, please explain how the carbon burial rates depend on temperature.

Page 30, line 22; page 31, lines 1-2: "Alkalinity flux from C1 exceeded that of other model versions towards the end of the simulation period as the slower carbon removal resulted in much warmer surface conditions compared to other model versions."

How does alkalinity flux relate to surface temperature in the model?

Pages 32-35: One important caveat that is not discussed here is the lack of nutrient

limitation in the UVic terrestrial module, which could have important implication for the prediction of primary production over long time scales and the resulting effect on modeled change in weathering.

———————————————

---

## Referee Comment (RC2) · Anonymous Referee #2 · 3 Feb 2017

This paper describes a terrestrial weathering scheme for the UVic model that takes into account the spatial variation of climate and rock types to calculate weathering rates. The reaction of the model to future warming is accessed. The study is one of only a few that look at the effect of using a 2D weathering scheme in climate models. The model seems appropriate for this study and the paper is appropriate for Earth System Dynamics. Overall I think the paper is interesting, well written and should be published after some relatively minor revisions.

—

General Comments:

A major weakness, in studies of long term weathering, lies in our poor understanding of the processes involved. While we may be able to roughly estimate current rates of weathering, future weathering rates are poorly constrained. The paper concludes that the effect of changes in weathering is small, on timescales less than 1000 years (consistent with other studies), and that changes in vegetation have a greater effect than changes in temperature and runoff. How robust are these conclusions? Perhaps it is impossible to know. This is not a criticism of the authors or the paper but an inevitable issue when modelling poorly understood processes. While the authors do recognize that the results are quite uncertain, it would be useful if they could at least try to quantify this uncertainty?

Specific Comments:

Page 9, Line 20: Given that you mention the GEOCLIM model is one of the few other models that have attempted a 2D weathering approach, it would be useful to briefly describe the differences between your implementation and what was done in GEOCLIM. You do provide some details with how 2D weathering was done in GENIE but not in the GEOCLIM (FOAM -LPJ) model.

Page 11, Line 22: It is actually the "net" sedimentation rate of CaCO3, which is the sedimentation rate minus the dissolution rate. You might want to call this the net burial rate to avoid confusion.

Page 14, Lines 8-9: If I am reading this correctly, it seems odd to me that you would have one scheme for steady state weathering, which depends only on runoff, area of a rock type and a constant weathering rate multiplier, while changes in weathering also depend on terrestrial biological production, a different form of runoff and temperature. Would it not be possible to derive global average steady state weathering for each rock type, using global average runoff (presumably this is what is done in a 0D model), and then apply different steady state weathering rates spatially, depending on deviations of

[Figure]

NPP, runoff and temperature from their global average values? From your description, I am assuming it is not done this way. Is there a reason not to? It seems more consistent to me. This would be a test of the robustness of how weathering rates change with (spatial) changes in climate. If you could still generate reasonable weathering rates spatially, it would help validate your parameterizations. This may be one of the only tests you can do. While it is not critical to change this, perhaps you could discuss this possibility.

Page 15, lines 14-17: In the version of the UVic model without interactive weathering, the global weathering rate of $CaCO_3$ during the spin-up is set to be equal to the global net sediment accumulation rate of $CaCO_3$ which ensures carbon conservation. The model spins up to a steady state within about 10,000 years. How was the spin-up done with the interactive weathering model? What is the net sediment accumulation rate compared to the overall weathering rate? It seems that either the global average weathering rate or the global average biological production that leads to net sedimentation of $CaCO_3$ should be adjusted so that they are equal under a specified level of $CO_2$. If not, then you will have drift in ocean carbon and alkalinity. If you are not doing this, how much drift in DIC and alkalinity do you see after your spin-up?

Page 21, line 1-2: It might be clearer if you say: "immediately balanced by an equivalent outgassing of carbon from the ocean". When you use "uptake" it sounds like you mean uptake by the ocean not the atmosphere. I realize you say uptake "from" the ocean so what you are saying is right - I just think it would be clearer to use the term outgassing.

Page 21, Lines 2-8: I am not sure I follow this. I would expect little delay - if you draw down a unit of $CO_2$ and send it to the ocean as DIC, the ocean should outgas a unit of $CO_2$ again, but if there is a significant delay in terms of alkalinity (as you suggest - although why is unclear), why not take $CO_2$ from the atmosphere, put it in the ocean and let $CO_2$ outgas naturally? Although I do not think this is necessary, it does seem more logical. Am I missing something?

Page 21, Lines 13-16: Why, if your weathering fluxes are steady, would it take so long to each equilibrium? Why is there any difference in equilibration time, compared to the default model? It is the slow reduction in CO2 that causes long equilibration with silicate weathering (given a CO2 perturbation), but if CO2 is fixed, why does this take so long? I am not sure I understand this. Is it really just that weathering does not equal sediment burial and any drift makes it look as if it is taking longer to equilibrate? If this is the case, then it will only reach "equilibrium" when you allow CO2 to change (changing the climate and thus weathering), and that would take a long time.

Page 22, Line 2: "Table 1" should be "Table 2"?

Page 24, Line 23: is the difference in CO2 between A0 and A2 really 164 ppmv at year 12000? It does not look like it in figure 3a – closer to zero maybe. Do you mean the 0D vs 2D was different by 164 ppmv? That would not be too surprising and not really comparable to Meissner et al. 2012.

Page 25, Line 10: "indifferent" seems a bit anthropomorphic - more poetic though.

Page 25, Lines 13-14: What do you mean here? The reaction of all the reservoirs seem different to me (and not unexpectedly). The land reservoir is behaving a bit like the atmosphere and the sediment a bit like the ocean. The land and sediment are certainly not behaving any more "differently" than the ocean and the atmosphere. If you mean the land and sediment react differently to large or extended emissions - that is not clear either. I must admit the sediment reaction does seem a bit odd. I would have expected the carbon content of sediment to decrease at some point (see next comment).

Page 25: Lines 18-20: It is not really clear that you are really plotting the change in CaCO3 mass in figure 3b. I suspect this is the change in total buried mass. This may be mislabeled in the model output. The buried mass of CaCO3 should be reducing for at least some of this time period but total mass (which includes clay) may well keep increasing. The reason I suspect this is the case, is that the change in the total mass

of CaCO3 should just be the integral of the difference between the accumulation and dissolution rates (or the integral of net burial). If you look at figure 6b or 8b, the change in dissolution rate is more than double the change in accumulation rate and the percent of CaCO3 in the pore layer heads lower after year 3000. This should mean that you have a negative change in total CaCO3 (which is what you would expect with carbonate compensation) and yet we do not see this in the change in buried mass. Even if the total buried mass of CaCO3 is still always increasing (burial rate is still positive), the slope must be decreasing after year 3000 since the change in dissolution is much greater than the change in accumulation. This is not obvious in the figure. I think your plots of CaCO3 buried mass need to be checked and possibly corrected.

Page 26, Lines 4-8: The pattern of warming looks pretty standard to me. Extra warming at the poles from polar amplification, more warming over land than ocean, more warming where vegetation has increased (including the tropics). Why does this need explanation? Why would static wind fields necessarily trap more heat in the tropics? Changes in wind fields could cause more divergence or more convergence of heat in the tropics. Atmospheric reorganization does not guarantee increased heat transport out of the tropics. Much of the heat transport in the UVic model is diffusive and that certainly does not trap heat in the tropics.

Page 26, Lines 20-22: Are you suggesting tropical forest die off in SE Asia and that they eventually grow back (by 12000 when NPP is similar again to PI)? Do other tropical forests show die off?

Page 27, Lines 2-3: What do you mean here? Do you mean statistically correlated? Can you state the strength of the correlations? Given that you have defined weathering to be basically linearly dependent on NPP but exponentially dependent on runoff, does comparing correlations (at least linear ones) mean anything?

Page 27, Lines 7-9: The significant changes in weathering near Kazakhstan and Zambia are curious. Nothing stands out in rock type, temperature, NPP or runoff that would

cause these large changes in weathering. Is this somehow an odd overlap of many factors? Something related to your complex NPP parameterization? Can you nail this down a bit?

Page 28, Lines 1-4: This leaves me wondering how robust these results are? Are these differences in dependences real or just due to uncertain parameter choices?

Page 28, Lines 9-15: I agree this is interesting but it would be more interesting if you knew why. Can this not be diagnosed from model output? Your explanation here is confusing. Is there a large decrease in vegetation NPP in A0 that would not affect B1 (or B2)? Is this from the drop off of NPP in SE Asia after year 3000 (as in Figure 4b)?

Page 29, Lines 9-12: This is the bit that I find really confusing. If the percent of CaCO3 in the pore layer is decreasing, then likely so should the total. This change in the total amount of CaCO3 should be checked against the integral of the net accumulation of CaCO3 (accumulation minus dissolution) at the sediment surface (see comment Page 25: Lines 18-20).

Page 29: Line 15: Should both "were"s be "are"s - given that we are not talking about figures in the past tense?

Page 29, Lines 20-22: I am assuming that you mean the pattern of weathering between B1 and B2 would look the same but the magnitude would be a bit higher in B1 (given that globally increased CO2 would increase weathering everywhere). Is that correct?

Page 31, Lines 20-22 and Page 32, Lines 1-2: I agree it must be deep alkalinity changes that are helping preserve the sediments and that both silicate and carbonate weathering are helping with this (comparing C1 or C2 to C3). Presumably silicate weathering (C2) is more effective at preserving sediments because it is also reducing DIC (compared to C1) and thus pCO2 and acidity. Would you agree?

Discussion: This section seems a bit speculative without more references.

Page 32, Lines 15,17: It seems to me that your parameterization of NPP is only quasi

linear and runoff is exponential, so I am not sure you can say that it varies linearly.

Page 33, Line 1: Do you mean "rely heavily on the ratio of NPP or runoff and their pre-industrial values" rather than "the ratio between initial weathering and initial NPP/runoff"?

Page 33, Lines 12-23: There is no doubt that the UVic model has a simplified atmosphere and this may limit a number of potential feedbacks. What is not clear, is how important these feedbacks are in the context of this paper. Expansion of atmospheric cells and poleward shifts in wind patterns with warming are, to a certain extent, captured by parameterizations in the model. It is not clear that the UVic model is overestimating tropical temperatures 1000 years into the future (as you also suggest in Section 3.1). What evidence do you have for this? Perhaps a more general statement about climate model uncertainty at these time scales would be more appropriate.

Page 34, Lines 1-2: Reference?

Page 34, Lines 5-19: Do you really think that sea level rise will inundate enough weathering active areas to make much of a difference. I am skeptical. Do you have any references?

Page 34, Line 20-21: The statement that "There has been an extensive discussion in recent years" is just begging for a reference.

Page 35, Lines 1-4: What processes? Reference?

Page 35, Lines 5-11: References?

Page 36, Lines 12-15: Again, I wonder how robust this is considering the uncertainty in the parameterizations.

Table 2: The caption mentions "pulses" even though I think few people would consider emissions spread over 1000 years to be a "pulse" (as in A2). I would just remove the references to pulses. Maybe something like: "The emission total is the total amount of

[Figure]

carbon emitted after year 2000, while the emission period represents the time span of emissions; the emission total is divided equally among the number of time steps during the emission period."

Figure 1: The caption for Figure 1 suggests that there are 3 panels (a), (b) and (c). Where is panel 1c? I think what is labeled as Figure 1b is really referred to as Figure 1c in the caption and the original Figure 1b was dropped. Is that correct? It seems reasonable not to include the original Figure 1b but the caption for Figure 1 should then be corrected.

Figure 2: The caption for Figure 2 refers to Figures 1b and Figures 1c. Figure 1c does not exist and it either needs to be included or the caption needs to be corrected.

Figure 3: In the caption for Figure 3b, maybe change "budgets" to anomalies or differences from pre-industrial. Budget is a bit ambiguous. Why do the lines for A0* and A1* stop before 12000 for DIC in Figure 3c but not the other panels?

Figures 3, 6 and 8: This is more a matter of personal style but I find it a bit distracting having the vertical axes labels switching from left to right on your line plots. I would think figures would be more compact and thus could be made larger if the axis were all on the left. The other thing that is a bit distracting is the change in horizontal scale which adds artificial breaks in the slopes of the lines. Would it be possible to show a small break in the line when you change scales just to emphasize that the slope is not really continuous? I wonder if this many scale changes are really necessary? The first vertical dashed line does not even seem to indicate a scale break and the last line is superfluous.

Figure 6: Again, in the caption for (b) maybe change "budget" to anomaly or difference from pre-industrial.

Figure 8: Change the caption for (b) as for Figure 6.
* * *

---

## Author Comment (AC1) · 14 Mar 2017

Authors' Answer to Review comments (answers are preceded by a dash "-")

Anonymous Referee #1

Brault et al. built spatially explicit parameterization schemes of continental weathering into the UVic Earth system model of intermediate complexity. Changes in weathering rate are parameterized based on rock types, changes in surface temperature, terrestrial primary production, river runoff, and atmospheric $CO_2$. Then the UVic model is used to project long-term changes in the global carbon cycle under assumed anthropogenic CO2 emission scenarios. Simulation results using spatially explicit weathering schemes are compared with those parameterizing weathering flux using global mean variables. In general, it is found that the terrestrial weathering has a negligible effect on the ocean biogeochemistry and climate change on the timescales from decades to centuries, but become much more important on multimillennial timescales. This finding is qualitatively in line with earlier modeling studies that incorporate a 0D weathering component.

The novelty of this study is the representation of terrestrial weathering on each model grid cell, which is an improvement over earlier studies using 0D weathering scheme. Given the large uncertainty of the weathering parameterization schemes (The authors did a nice job in discussing those uncertainties, though) and the model-simulated changes in relevant variables over the timescale of many thousands of years, it is difficult to make an assessment of how reliable the quantitative results are. Nevertheless, this work is one of the few Earth system modeling studies that provides a long-term projection of the global carbon cycle and climate with a spatially explicit weathering component. This manuscript is well written. I recommend its publication with a few minor comments.

Specific comments:

The six-page introduction is a comprehensive review of the rock weathering processes, in particular the modeling studies of weathering effect on the carbon cycle. This kind of introduction is useful for broad readers to understand the weathering effect. However, this type of introduction might be too lengthy to fit the journal. The authors may need to condense it.

- Before proceeding with any condensing of the introductory paragraphs, we would like to know whether the ESD editor agrees with the above statement. If the introduction is indeed too lengthy for this journal, we will shorten sections 1.3, 1.4, and 1.5 accordingly.

Page 3, lines 10-11: "However, there have been but very few quantitative assessments

of its impacts on carbon cycling and ocean biogeochemistry.." ; "very few quantitative assessment" is not an objective statement. There are actually quite a number of existing modeling studies, as cited in the following section, on the effect of rock weathering on the carbon cycle.

- The sentence in question will be modified as follows to remove the subjective statement: "However, its relevance over timescales [. . .] is largely unknown."

Page 11, lines 4-5 "The UVic ESCM also includes a fully coupled global carbon cycle, which consists of inorganic carbon chemistry and land-surface exchanges of CO2 (Ewen et al., 2004)"; "land-surface exchanges of CO2" should be "air-sea exchange of CO2"

- The sentence will be modified as suggested.

Page 15, lines 16-17: "and on par with previous estimations of pre-industrial global weathering intensity." Some references should be given here for the data-based estimate of pre-industrial global weathering intensity.

- Many references will be added to the paper to address the issue of missing references that was raised more than once in these reviews. In the present case, the authors will be citing Moon et al. [2014], Geochim. Cosmochim. Acta.

Page 19, lines 14-15: "Weathering is calculated in each individual land grid cell, and routed to the coastal ocean as fluxes of alkalinity and dissolved inorganic carbon"; It would be helpful to elaborate a bit more of how the river routing is done in the UVic model.

- We did not believe it useful to include such information in the manuscript. The Weaver et al. [2001] paper which introduces the UVic model does a good job in describing multiple aspects of the model, such as the distribution of catchment basins and how river runoff is evenly distributed along the coastal margins of the drainage area it is associated with. We would kindly request the editor's opinion on the matter.

Page 21, " Pre-industrial steady-state weathering was obtained by integrating the model for over 20,000 years under year 1800 boundary conditions ... However, the fixing of deep ocean alkalinity and dissolved inorganic carbon (DIC) content would have required as much as 10ˆ5 model years – an impossibly long simulation time given the level of complexity of the UVic model."; It would be useful to show something like time series of ocean-mean DIC and alkalinity to see how far the system is away from equilibrium after 20,000 years of spinup.

- Spinup runs are rarely discussed in climate model papers as they are not that interesting. Such timeseries as suggested by the reviewer may be added to the manuscript, but we do not believe it would provide very useful information. Perhaps we can include an additional sentence discussing the equilibrium of the ocean system with regards to global mean DIC and alkalinity contents, and stress this as a potential source of uncertainty in the discussion section.

Page 25, lines 22-23, please explain how the carbon burial rates depend on temperature.

- By "carbon burial rates" we effectively refer to the precipitation rate of calcium carbonate, and the solubility of the latter in seawater increases with increasing water temperature, as is the case for most chemical solutions. We will add this short explanation to the manuscript.

Page 30, line 22; page 31, lines 1-2: "Alkalinity flux from C1 exceeded that of other model versions towards the end of the simulation period as the slower carbon removal resulted in much warmer surface conditions compared to other model versions."; How does alkalinity flux relate to surface temperature in the model?

- Perhaps the reviewer was confused by our usage of "alkalinity flux" to refer to the influx of alkalinity due to terrestrial weathering? Alkalinity flux is calculated from carbonate and silicate weathering flux according to equation 18, with the respective temperature dependence for both types of weathering given by equations 7 and 8.

[Figure]

Pages 32-35: One important caveat that is not discussed here is the lack of nutrient limitation in the UVic terrestrial module, which could have important implication for the prediction of primary production over long time scales and the resulting effect on modeled change in weathering.

- The reviewer raises an excellent point that we had not thought about. Such a shortcoming may lead to some uncertainties in the results. We shall add a few sentences on this in the discussion section.

Anonymous Referee #2

This paper describes a terrestrial weathering scheme for the UVic model that takes into account the spatial variation of climate and rock types to calculate weathering rates. The reaction of the model to future warming is accessed. The study is one of only a few that looks at the effect of using a 2D weathering scheme in climate models. The model seems appropriate for this study and the paper is appropriate for Earth System Dynamics. Overall I think the paper is interesting, well written and should be published after some relatively minor revisions.

General Comments:

A major weakness, in studies of long term weathering, lies in our poor understanding of the processes involved. While we may be able to roughly estimate current rates of weathering, future weathering rates are poorly constrained. The paper concludes that the effect of changes in weathering is small, on timescales less than 1000 years (consistent with other studies), and that changes in vegetation have a greater effect than changes in temperature and runoff. How robust are these conclusions? Perhaps it is impossible to know. This is not a criticism of the authors or the paper but an inevitable issue when modelling poorly understood processes. While the authors do recognize that the results are quite uncertain, it would be useful if they could at least try to quantify this uncertainty?

[Figure]

- We completely agree with the reviewer that the effect of changes in vegetation on weathering is subject to much uncertainty. The issue of how biological enhancements of weathering should be represented through changes in vegetation NPP is a daunting task indeed, and unfortunately neither our understanding of the underlying processes nor the level of complexity in the UVic model terrestrial module allow for a parameterization of the biological enhancement that would reflect the underlying physical mechanisms. At this point it would be very difficult to quantify the uncertainty related to our choice of NPP parameterization, as there is no truly alternative formulation of the biological enhancement available in the literature to this day with which we could compare our results. The only other option is to revert to CO2 as the main proxy for biological activity, which is arguably a worse choice yet than NPP.

Specific Comments:

Page 9, Line 20: Given that you mention the GEOCLIM model is one of the few other models that have attempted a 2D weathering approach, it would be useful to briefly describe the differences between your implementation and what was done in GEOCLIM. You do provide some details with how 2D weathering was done in GENIE but not in the GEOCLIM (FOAM -LPJ) model.

- We will include a short description of the GEOCLIM approach to weathering parameterization. Note that GEOCLIM is a much coarser resolution model and thus there are few similarities with the UVic model.

Page 11, Line 22: It is actually the "net" sedimentation rate of CaCO3, which is the sedimentation rate minus the dissolution rate. You might want to call this the net burial rate to avoid confusion.

- We agree that such a specification would reduce confusion. It will be changed in the future version of the manuscript.

Page 14, Lines 8-9: If I am reading this correctly, it seems odd to me that you would

have one scheme for steady state weathering, which depends only on runoff, area of a rock type and a constant weathering rate multiplier, while changes in weathering also depend on terrestrial biological production, a different form of runoff and temperature. Would it not be possible to derive global average steady state weathering for each rock type, using global average runoff (presumably this is what is done in a 0D model), and then apply different steady state weathering rates spatially, depending on deviations of NPP, runoff and temperature from their global average values? From your description, I am assuming it is not done this way. Is there a reason not to? It seems more consistent to me. This would be a test of the robustness of how weathering rates change with (spatial) changes in climate. If you could still generate reasonable weathering rates spatially, it would help validate your parameterizations. This may be one of the only tests you can do. While it is not critical to change this, perhaps you could discuss this possibility.

- This is a very interesting comment. The advantage of 0-D models first-hand is that steady state weathering does not need to be calculated; a value for pre-industrial global weathering fluxes can be obtained from measurements, or by setting it to equal pre-industrial net sedimentation rate (in other words, the 0D model does not need to take into account runoff, or any other model parameter). For the 2-D model, the only variable for which we believe this approach would make a difference is NPP, and this is already somewhat accounted for in our new parameterization of NPP (see equation 10). Still, it would be interesting to test an approach as suggested by the reviewer and compare it with current model results. However, we believe this is beyond the present scope of the paper.

Page 15, lines 14-17: In the version of the UVic model without interactive weathering, the global weathering rate of CaCO3 during the spin-up is set to be equal to the global net sediment accumulation rate of CaCO3 which ensures carbon conservation. The model spins up to a steady state within about 10,000 years. How was the spin-up done with the interactive weathering model? What is the net sediment accumulation

rate compared to the overall weathering rate? It seems that either the global average weathering rate or the global average biological production that leads to net sedimentation of CaCO3 should be adjusted so that they are equal under a specified level of CO2. If not, then you will have drift in ocean carbon and alkalinity. If you are not doing this, how much drift in DIC and alkalinity do you see after your spin-up?

- The interactive weathering model (equations 14-15) was not used when spinning up the model. Instead steady-state weathering rates were calculated using only equations 5-6, thus only taking into account changes in runoff (which were quite small). The reasoning behind our not including temperature or NPP in the spinup is that the steady-state weathering rates, which depend on the boundary conditions relevant to preindustrial conditions (including atmospheric CO2 concentrations), must necessarily reflect the steady-state climate and environment. Given that this method produced a global weathering flux very similar to the one used in the 0-D version of the model, it made sense to only turn on the interactive weathering scheme when doing transient model runs.

Page 21, line 1-2: It might be clearer if you say: "immediately balanced by an equivalent outgassing of carbon from the ocean". When you use "uptake" it sounds like you mean uptake by the ocean not the atmosphere. I realize you say uptake "from" the ocean so what you are saying is right - I just think it would be clearer to use the term outgassing.

- We will make the change as suggested in the next version of the manuscript.

Page 21, Lines 2-8: I am not sure I follow this. I would expect little delay - if you draw down a unit of CO2 and send it to the ocean as DIC, the ocean should outgas a unit of CO2 again, but if there is a significant delay in terms of alkalinity (as you suggest - although why is unclear), why not take CO2 from the atmosphere, put it in the ocean and let CO2 outgas naturally? Although I do not think this is necessary, it does seem more logical. Am I missing something?

- It is far simpler to represent the effect of weathering in terms of fluxes of DIC and alkalinity to the ocean rather than tamper with atmospheric CO2 content. Effectively, the short-circuiting of the atmosphere is used in order to avoid the unnecessary complication of having to remove CO2 directly from the atmosphere. The delay which we allude to in the paper comes from the average time it takes for the calcium and carbonate ions in the seawater to precipitate again as calcium carbonate, releasing a molecule of carbon dioxide. This delay would be expected to be on the same order of length as the typical mixing timescale of the ocean, with is 103-104 years.

Page 21, Lines 13-16: Why, if your weathering fluxes are steady, would it take so long to reach equilibrium? Why is there any difference in equilibration time, compared to the default model? It is the slow reduction in CO2 that causes long equilibration with silicate weathering (given a CO2 perturbation), but if CO2 is fixed, why does this take so long? I am not sure I understand this. Is it really just that weathering does not equal sediment burial and any drift makes it look as if it is taking longer to equilibrate? If this is the case, then it will only reach "equilibrium" when you allow CO2 to change (changing the climate and thus weathering), and that would take a long time.

- The main reason the model takes so long to equilibrate is that there was no steady-state equilibrium of the current version of the model available at year 1800; thus we had to take a transient snapshot of the model at year 1800 and equilibrate it at year 1800 conditions along with the non-constant weathering scheme, which means ocean DIC and alkalinity were not necessarily at their steady-state values. Due to the long response time of ocean with regard to these two parameters, it is expected that any steady-state equilibrium run would last well beyond 104 model years.

Page 22, Line 2: "Table 1" should be "Table 2"?

- Yes; we will make the changes as suggested.

Page 24, Line 23: is the difference in CO2 between A0 and A2 really 164 ppmv at year 12000? It does not look like it in figure 3a – closer to zero maybe. Do you mean the 0D vs 2D was different by 164 ppmv? That would not be too surprising and not really

comparable to Meissner et al. 2012.

- The reviewer is correct. . . the value of 164ppm arises from comparing the final CO2 concentration of A0 between its 2D and 0D model versions, which is an unintended mistake. In truth, the difference between A0 and A2 is a much more modest 13ppm (8ppm when comparing the two 0D model runs). We shall rewrite the beginning of the paragraph accordingly, noting that the very small end-of-run difference between A0 and A2 supports very well our conclusion that the duration of the carbon emission does not impact the long-term perturbation to the Earth system; only the total amount of carbon does, as well as the strength of the negative feedback mechanism which involves weathering.

Page 25, Line 10: "indifferent" seems a bit anthropomorphic - more poetic though.

- We shall leave the text as it is.

Page 25, Lines 13-14: What do you mean here? The reactions of all the reservoirs seem different to me (and not unexpectedly). The land reservoir is behaving a bit like the atmosphere and the sediment a bit like the ocean. The land and sediment are certainly not behaving any more "differently" than the ocean and the atmosphere. If you mean the land and sediment react differently to large or extended emissions – that is not clear either. I must admit the sediment reaction does seem a bit odd. I would have expected the carbon content of sediment to decrease at some point (see next comment).

- Here we refer to the difference in behavior between the 0D and 2D curves for the land and sediment reservoirs, which is much more pronounced than for the atmospheric and oceanic reservoirs. These differences in behavior are then discussed below. So saying that "land and sediment react differently to large or extended emissions" would be the more accurate way of describing what we are discussing here. The contents of lines 13-14 will be altered slightly to reduce any potential confusion over this.

Page 25: Lines 18-20: It is not really clear that you are really plotting the change in CaCO3 mass in figure 3b. I suspect this is the change in total buried mass. This may be mislabeled in the model output. The buried mass of CaCO3 should be reducing for at least some of this time period but total mass (which includes clay) may well keep increasing. The reason I suspect this is the case, is that the change in the total mass of CaCO3 should just be the integral of the difference between the accumulation and dissolution rates (or the integral of net burial). If you look at figure 6b or 8b, the change in dissolution rate is more than double the change in accumulation rate and the percent of CaCO3 in the pore layer heads lower after year 3000. This should mean that you have a negative change in total CaCO3 (which is what you would expect with carbonate compensation) and yet we do not see this in the change in buried mass. Even if the total buried mass of CaCO3 is still always increasing (burial rate is still positive), the slope must be decreasing after year 3000 since the change in dissolution is much greater than the change in accumulation. This is not obvious in the figure. I think your plots of CaCO3 buried mass need to be checked and possibly corrected.

- We are indeed plotting the total buried mass here, which is a means to quantify the total amount of carbon present in the sediment reservoir. We believe this graph should remain as it is, but it will be renamed as "sediment carbon content" to avoid the confusion generated here. However, the CaCO3 pore layer mass does display a behavior much like what the reviewer describes: steady between years 2000-3000, then a sharp drop between years 3000-6000 followed by a slow recovery. Such results would be useful to show in order to better discuss changes in ocean carbonate chemistry in our model runs. We shall include this graph within figure 3 and add the relevant discussion to the manuscript.

Page 26, Lines 4-8: The pattern of warming looks pretty standard to me. Extra warming at the poles from polar amplification, more warming over land than ocean, more warming where vegetation has increased (including the tropics). Why does this need explanation? Why would static wind fields necessarily trap more heat in the tropics?

Changes in wind fields could cause more divergence or more convergence of heat in the tropics. Atmospheric reorganization does not guarantee increased heat transport out of the tropics. Much of the heat transport in the UVic model is diffusive and that certainly does not trap heat in the tropics.

- While some of the tropical warming can be explained by the vegetation change, it was unexpected that some tropical regions would warm as much as polar latitudes – in general, paleoclimate studies indicate that in a warmer world the tropics would only be marginally warmer than today. It is true that we cannot predict with the UVic model how this global warming would translate into changes in atmospheric circulation, but records from the past indicate that an increase in global temperatures almost always translate into an increased capacity of the Earth system to transport heat from the tropics poleward. We do not believe necessary to alter the discussion in lines 4-8.

Page 26, Lines 20-22: Are you suggesting tropical forest die off in SE Asia and that they eventually grow back (by 12000 when NPP is similar again to PI)? Do other tropical forests show die off?

- The coupled TRIFFID-MOSES2 indeed suggests that as SE Asian temperatures drastically warm by year 3000, C4 grasses eventually replace broadleaved trees as the dominant plant functional type. The validity of this outcome can be questioned, just as we question the validity of the 9°C surface air temperature increase over the region which is the apparent cause of this shift in vegetation regime. It is possible that the tropical forests would attempt to resist the extreme warming through increased evapotranspiration rates, for example, to avoid being exposed to temperatures that would be threatening to their survival. It is also possible that due to the short time scale of the perturbation (a few thousand years), plant species would not have time to adapt to the rapid warming and would indeed die off and be replaced by a better suited plant functional type – this could help explain the extremely high temperature increase over the region. Other tropical forests do not, in fact, show this behavior. The Amazonian forest, in particular, remains remarkably stable over the same time period, with much

smaller changes in NPP and a smaller increase in surface air temperature over the region as well.

Page 27, Lines 2-3: What do you mean here? Do you mean statistically correlated? Can you state the strength of the correlations? Given that you have defined weathering to be basically linearly dependent on NPP but exponentially dependent on runoff, does comparing correlations (at least linear ones) mean anything?

- By "correlation" here, we meant only a qualitative visual assessment (the weathering curve most strongly resembles the NPP curve, which implies that NPP is the most significant factor for weathering in our model output) which supports our conclusion that weathering is most affected by changes in NPP. If we were to go further (i.e., "comparing correlations") we would compare the relative importance of each of the parameters (temperature, NPP, runoff) in equations 10-11, but we do not believe it necessary to do such an analysis.

Page 27, Lines 7-9: The significant changes in weathering near Kazakhstan and Zambia are curious. Nothing stands out in rock type, temperature, NPP or runoff that would cause these large changes in weathering. Is this somehow an odd overlap of many factors? Something related to your complex NPP parameterization? Can you nail this down a bit?

- The changes over Zambia make sense as this region is much more dense in carbonate rocks than elsewhere in tropical Africa. As for Kazakhstan, however, the output underlines an artifact of our complex NPP parameterization. In short, although our parameterization mostly allows us to eliminate large weathering changes in region of low initial NPP, it does not succeed to do so in central Asia where just the right combination of low initial NPP and a high enough initial weathering rate makes it possible for weathering values to become excessively high when the increase in temperature results in a drastic (and very temporary) increase in vegetation productivity. We shall add a few sentences to the discussion on this.

Page 28, Lines 1-4: This leaves me wondering how robust these results are? Are these differences in dependences real or just due to uncertain parameter choices?

- The differences in dependences are very real as they are caused by deliberate changes in the parameterizations. We are comparing here three radically different interpretations of the biological enhancement of weathering (NPP vs. CO2 vs. none), and the model output is meant to display just how much these differences mean in terms of quantifying the changes in terrestrial weathering.

Page 28, Lines 9-15: I agree this is interesting but it would be more interesting if you knew why. Can this not be diagnosed from model output? Your explanation here is confusing. Is there a large decrease in vegetation NPP in A0 that would not affect B1 (or B2)? Is this from the drop off of NPP in SE Asia after year 3000 (as in Figure 4b)?

- This could be diagnosed from model output by isolating all of the factors which affect weathering rates but such an analysis is beyond the scope of this paper. The most logical answer to why the weathering rates in A0 drop below those of B1 is that the temperature (and NPP) in A0 have dropped so much below that of B1 that weathering rates in B1 manage to exceed those in A0 despite using a scheme which does not as efficiently represent the strength of the biological enhancement of weathering.

Page 29, Lines 9-12: This is the bit that I find really confusing. If the percent of CaCO3 in the pore layer is decreasing, then likely so should the total. This change in the total amount of CaCO3 should be checked against the integral of the net accumulation of CaCO3 (accumulation minus dissolution) at the sediment surface (see comment Page 25: Lines 18-20).

- The total amount of carbon contained within the sediment reservoir increases; however the amount of precipitated CaCO3 fluctuates according to the chemistry of seawater. In the text we shall replace "CaCOň3 buried mass" with "carbon budget of sediments" to avoid confusion. Along with the changes in page 25, this should hopefully clear the confusion around our interpretation of results pertaining to the sediments.

Page 29: Line 15: Should both "were"s be "are"s - given that we are not talking about figures in the past tense?

- The changes will be made as suggested.

Page 29, Lines 20-22: I am assuming that you mean the pattern of weathering between B1 and B2 would look the same but the magnitude would be a bit higher in B1 (given that globally increased CO2 would increase weathering everywhere). Is that correct?

- Yes that is correct. The text will be slightly modified so that no such assumptions are required from the reader.

Page 31, Lines 20-22 and Page 32, Lines 1-2: I agree it must be deep alkalinity changes that are helping preserve the sediments and that both silicate and carbonate weathering are helping with this (comparing C1 or C2 to C3). Presumably silicate weathering (C2) is more effective at preserving sediments because it is also reducing DIC (compared to C1) and thus pCO2 and acidity. Would you agree?

- We agree that silicate weathering is in general much more efficient at dealing with changes in atmospheric CO2 compared with carbonate weathering. In terms of how this is represented in our model, silicate weathering increases the ratio of alkalinity flux to DIC flux, with effects on oceanic pCO2 and alkalinity as outlined by the reviewer. On top of helping with sediment preservation (same as carbonate weathering), the decrease in DIC results in a permanent transfer of carbon from the atmosphere to the ocean.

Discussion: This section seems a bit speculative without more references.

Page 32, Lines 15,17: It seems to me that your parameterization of NPP is only quasi linear and runoff is exponential, so I am not sure you can say that it varies linearly.

- This does create some confusion. The sentence will be modified so that the variation will be described only as monotonic.

Page 33, Line 1: Do you mean "rely heavily on the ratio of NPP or runoff and their pre-industrial values" rather than "the ratio between initial weathering and initial NPP/runoff"?

- We mean both, but especially the ratio between initial weathering and initial NPP/runoff. If initial weathering is (relatively) high but initial NPP/runoff is low, then any increase in the latter would result in a disproportionately large increase of the former. We shall slightly modify the lines to clear up any confusion.

Page 33, Lines 12-23: There is no doubt that the UVic model has a simplified atmosphere and this may limit a number of potential feedbacks. What is not clear, is how important these feedbacks are in the context of this paper. Expansion of atmospheric cells and poleward shifts in wind patterns with warming are, to a certain extent, captured by parameterizations in the model. It is not clear that the UVic model is overestimating tropical temperatures 1000 years into the future (as you also suggest in Section 3.1). What evidence do you have for this? Perhaps a more general statement about climate model uncertainty at these time scales would be more appropriate.

- The evidence comes from comparing the UVic model output with other similar experiments. For example, Clark et al. [2016] (Nature climate change) simulate the multi-millennial evolution of surface air temperatures and sea level change following various pulse emissions of carbon at year 2000, the highest of which is slightly above 5000 Pg. Their surface maps do not show any trace of the intense tropical warming that we obtain with the UVic model. That being said, we shall include a general statement about climate model uncertainty at these time scales.

Page 34, Lines 1-2: Reference?

- Any of the early papers on weathering could be cited. For example, Walker and Kasting [1981], Berner [1991], Lenton and Britton [2006].

Page 34, Lines 5-19: Do you really think that sea level rise will inundate enough weathering active areas to make much of a difference. I am skeptical. Do you have any references?

- Clark et al. [2016] project a sea level rise of more than 50m in their "business as usual" scenario. This would be more than enough to flood many coastal areas and have a more than trivial impact on weathering rates, according to us. Of course, this could only be verified by applying a 2-D model to the question.

Page 34, Line 20-21: The statement that "There has been an extensive discussion in recent years" is just begging for a reference.

- We shall add references to support this statement.

Page 35, Lines 1-4: What processes? Reference?

- These processes could range from anywhere between the physical breakdown and grinding of rocks by roots to the chemical enhancement of weathering due to the presence of active reagents. Again references will be added to support this statement.

Page 35, Lines 5-11: References?

- This paragraph was meant more as a general discussion of potential factors which may or may not impact weathering, and thus suggestions for future research.

Page 36, Lines 12-15: Again, I wonder how robust this is considering the uncertainty in the parameterizations.

- The robustness of this conclusion is discussed extensively in sections 3 and 4. According to the model versions used in the present paper (which are derived from the most recent interpretations of environmental impacts on weathering rates), the biological enhancement of weathering is the most important factor, and parameterizing it according to NPP yields a much larger response compared to other parameterizations. This finding is supported by other studies which have used NPP as a determining factor (ex. Lenton and Britton [2006], Meissner et al. [2012]).

Table 2: The caption mentions "pulses" even though I think few people would consider emissions spread over 1000 years to be a "pulse" (as in A2). I would just remove the references to pulses. Maybe something like: "The emission total is the total amount of emissions; the emission total is divided equally among the number of time steps during the emission period."

- The table caption will be modified as suggested by the reviewer.

Figure 1: The caption for Figure 1 suggests that there are 3 panels (a), (b) and (c). Where is panel 1c? I think what is labeled as Figure 1b is really referred to as Figure 1c in the caption and the original Figure 1b was dropped. Is that correct? It seems reasonable not to include the original Figure 1b but the caption for Figure 1 should then be corrected.

- This is an error carried from earlier versions of the manuscript. Figure 1 should only have two panels: (a) the adaptation to the UVic model resolution, and (b) the interpolated rock type fraction in each grid cell. This will be modified in the next manuscript version.

Figure 2: The caption for Figure 2 refers to Figures 1b and Figures 1c. Figure 1c does not exist and it either needs to be included or the caption needs to be corrected.

- See reply to the comment immediately above.

Figure 3: In the caption for Figure 3b, maybe change "budgets" to anomalies or differences from pre-industrial. Budget is a bit ambiguous. Why do the lines for A0* and A1* stop before 12000 for DIC in Figure 3c but not the other panels?

- We shall make the changes as suggested by the reviewer (we will use "differences"). The lines for A0* and A1* disappear in figure 3c because they exceed the maximum range set for the graph. In the next manuscript version, this figure will be redrawn with a higher maximum range for this panel, allowing us to see the entire curves.

Figures 3, 6 and 8: This is more a matter of personal style but I find it a bit distracting

having the vertical axes labels switching from left to right on your line plots. I would think figures would be more compact and thus could be made larger if the axis were all on the left. The other thing that is a bit distracting is the change in horizontal scale which adds artificial breaks in the slopes of the lines. Would it be possible to show a small break in the line when you change scales just to emphasize that the slope is not really continuous? I wonder if this many scale changes are really necessary? The first vertical dashed line does not even seem to indicate a scale break and the last line is superfluous.

- The reason that the vertical axes switch from left to right is to avoid cluttering all of the text and numbers on one side, which also allows us to make the text larger. We would argue that the time scale changes best reflect the evolution of the model output on various timescales. Note that the vertical dashed lines do not necessarily indicate scale breaks, but actually denote the times for which results are shown in figures 4, 5, and 7. The figure captions will be modified to indicate that.

Figure 6: Again, in the caption for (b) maybe change "budget" to anomaly or difference from pre-industrial.

- Changes will be made as suggested (we will use "differences").

Figure 8: Change the caption for (b) as for Figure 6.

- Same as above.

---

## Author Response (AR1)

**List of Relevant Changes**

- Decreased the length of the introduction. Sections 1.3 and 1.4 were merged together, and about half the length of the new section 1.4 as well as 1.5 has been removed.
- Added a short explanation of river routing at the location identified by Reviewer #1.
- Added a sentence to describe how close to equilibrium our spinup run was after 10,000 model years. We do not believe it relevant to include a timeseries of the model reaching steady-state.
- Added a paragraph in section 4 to further discuss the uncertainty relative to the formulation of NPP.
- Added a sentence to better explain GEOCLIM at the location identified by Reviewer #2.
- Added a sentence to address Reviewer comment at page 15, line 14-17.
- Added a few sentences to address Reviewer comment at page 21, line 2-8.
- Added a sentence to address Reviewer comment at page 21, line 13-16.
- Added a sentence to address Reviewer comment at page 24, line 23.
- Replaced "CaCO3 buried mass" with "sediment carbon budget" in figures 3b, 6b, and 8b.
- Added paragraph to address Reviewer comment at page 26, lines 20-22.
- Added a sentence to address Reviewer comment at page 27, line 2-3.
- Added a sentence to address Reviewer comment at page 28, line 1-4.
- Added a sentence to address Reviewer comment at page 28, line 9-12.
- Added reference to address Reviewer comment at page 35, line 1-4.
- Some of the other recommended changes by the reviewers (but not indicated by the editor) have also been addressed.